Manuscript prepared for Atmos. Chem. Phys.
with version 2015/09/17 7.94 Copernicus papers of the LaTeX class coperni-
cus.cls.
Date: 28 March 2017

# A new downscaling method for sub-grid turbulence modeling

**L. Rottner[1], C. Baehr[1], F. Couvreux[1], G. Canut[1], and T. Rieutord[1]**

[1]Météo-France - CNRS, CNRM / GAME, UMR 3589, 42 avenue Coriolis, 31100 Toulouse

*Correspondence to:* Lucie Rottner (lucie.rottner@meteo.fr)

**Abstract.**

In this study we explore a new way to model sub-grid tur-
bulence using particle systems. The ability of particle sys-
tems to model small-scale turbulence is evaluated using high-
resolution numerical simulations. These high-resolution data
are averaged to produce a coarse-grid velocity field which is
then used to drive a complete particle-system-based down-
scaling. Wind fluctuations and turbulent kinetic energy are
compared between the particle simulations and the high-
resolution simulation. Despite the simplicity of the physical
model used to drive the particles, the results show that parti-
cle system is able to represent the average field. It is shown
that this system is able to reproduce much finer turbulent
structures than the numerical high-resolution simulations. In
addition, this study provides an estimate of the effective spa-
tial and temporal resolution of the numerical models. This
highlights the need for higher resolution simulations in or-
der to evaluate the very-fine turbulent structures predicted by
the particle systems. Finally, a study of the influence of the
forcing scale on the particle system is presented.

## 1 Introduction

Following the increase in computing power, the resolutions
of meteorological models have increased steadily over the
past years. The refinement of the temporal and spatial reso-
lution of atmospheric models requires a finer and finer repre-
sentation of physical phenomena. The current weather fore-
cast models have resolutions of approximately one kilometer.
However, the small processes, which have local effects, are
still sub-grid processes in such models. Thus, they are subject
to physical parametrization.

The issue of downscaling concerns many meteorological
research fields, from snow pack modeling to cloud-cover
modeling. A particularly delicate matter is the modeling of
the turbulence in the Atmospheric Boundary Layer (ABL).
In the ABL, there is a transfer of energy from scales of the
order of a kilometer down to sub-meter scales. This transfer
is called the energy cascade. Thus, whatever the model reso-
lution, some turbulent processes are sub-grid processes. For
numerical weather forecast models, the processes associated
to sub-kilometer scales are not resolved yet. For instance, a
recent study shows that these processes are not resolved at
the scale of AROME Airport which has a horizontal resolu-
tion of 500 meters (Hagelin et al., 2014). Since turbulence
is a key driver of the evolution of local-scale atmosphere,
it is critical that turbulence processes are parametrized well.
For instance, recent studies have shown the influence of the
turbulence parametrization on the cloud modeling in tropical
regions (Machado and Chaboureau, 2014). Several field ex-
periments have helped to understand the influence of small-
scale turbulence on local weather conditions – the erosion
of the nocturnal valley inversion for instance (Rotach et al.,
2004; Drobinski et al., 2007; Rotach et al., 2008).

Because of their variability and their sensitivity to local
conditions, these turbulent phenomena are especially diffi-
cult to model. Instead of a reduction in grid size, we suggest
here another way to model sub-grid turbulence. In this paper,
we present a stochastic downscaling approach. Our method
is based on particle systems that are driven by a local turbu-
lence model. Those particles are embedded in grid cells (il-
lustration 1). From the mathematical point of view, the parti-
cles sample the probability density function (pdf) of the sub-
grid wind. The description of sub-grid processes based on
their pdf has been introduced by Sommeria and Deardorff
since 1977 (Sommeria and Deardorff, 1977). The Gaussian
approximation they made to describe the liquid water con-
tent has then been extended to other variables. Nowadays,
this kind of approximation is still widely used for downscal-
ing (Larson et al., 2002; Perraud et al., 2011; Larson et al.,
2012; Jam et al., 2013; Bogenschutz and Krueger, 2013).
The method we suggest differs from these previous works
in that the Gaussian assumption of the pdf shape is only lo-
cally made. This locally-Gaussian assumption is linked to
the use of a local average operator presented in section 3.5.

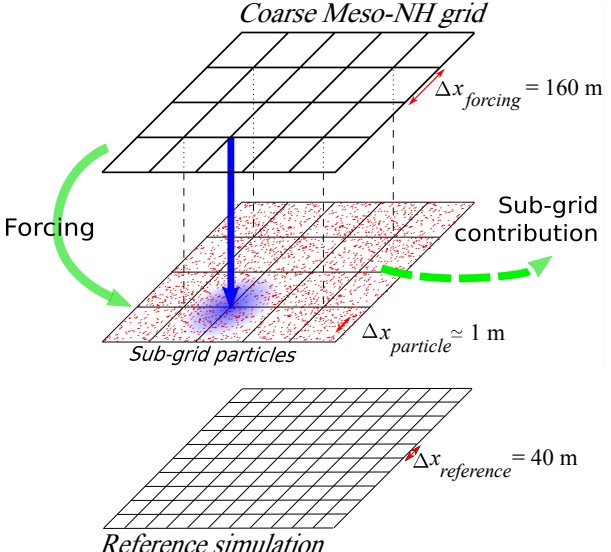

**Figure 1.** The downscaling scheme. Coarse fields are used to force a sub-grid particle system. The sub-grid fields are compared to a reference Meso-NH simulation.

The locally-Gaussian assumption is not equivalent to having a Gaussian pdf in each cell. Rather, in a given grid cell, particles are sampled from different Gaussian pdfs. Therefore, they yield a discrete pdf which is not necessarily Gaussian. The pdf time evolution is thus given by the particle evolution. This proposed particle approach facilitates the modeling of physical phenomena with nonlinear temporal evolutions. However, depending on the particle model, particle methods may have drawbacks such as interpolation issues, for instance as discussed in Brackbill et al. (1988). In the present work, the only delicate point is to ensure that the particle density is high enough in each grid cell.

In order to keep the average particles behavior consistent with the grid-point model, some grid-point fields are used as an external forcing on the particle system. The grid-point fields provide the values of the control parameters of the particle evolution model. This forcing is constant during the grid-point model time step and is applied every time new values are available. However, the particle evolution is performed at a shorter time step. Thus, the suggested downscaling method enables the refinement of both time and space scales.

In this work, the French research model Meso-NH is used to obtain high-resolution grid-point fields. The chosen simulations have been performed for the BLLAST experiment (Lothon et al., 2014). Therefore, we have used simulations and observations of real turbulent ABL to develop a stochastic downscaling method suitable for a limited area model.

First, the framework is presented. The BLLAST field experiment and the particle system and grid-point model coupling scheme are introduced. A description of the models fol-

lows in the section 3. Then, section 4.2 details the forcing procedure. A brief presentation of the turbulent kinetic energy computation is given in section 5. The results obtained using the suggested downscaling method are then presented. We finally discuss their sensitivity to the resolution of the forcing fields in section 7.

## 2 The BLLAST experiment

The BLLAST (Boundary Layer Late Afternoon and Sunset Turbulence) field campaign was conducted from 14 June to 8 July 2011 in southern France, in an area of complex and heterogeneous terrain. The BLLAST experiment resulted from a collaboration of several European laboratories spearheaded by the Laboratoire d'Aérologie. The experiment aim is to study the turbulence in the boundary layer during the late afternoon transition (Lothon et al., 2014).

To perform this study, all turbulence sources were investigated. A wide range of integrated instrument platforms including full-size aircraft, remotely piloted aircraft systems (RPAS), remote sensing instruments, radiosoundings, tethered balloons, surface flux stations, and various meteorological towers were deployed over different surface types (Pardyjak et al., 2011). In addition to the numerous observations, high-resolution simulations of the boundary layer have been done using Large-Eddy Simulation (LES) models. The ability of different meso-scale models to simulate turbulence has also been evaluated (Jimenez et al., 2014).

The BLLAST experiment addresses a wide range of scientific issues such as the turbulence decrease (Darbieu et al., 2015), the wind direction variability, or the turbulent kinetic energy budget (Nilsson et al., 2016). The diversity of the available observations and simulations, and the dynamism of the BLLAST community lead us to choose this experiment to develop the presented downscaling method.

## 3 Models

In this section, we introduce the two models used in this work. First, the grid-point model is presented, and the coupling scheme is described. Then, the focus is put on the particle system and its evolution model.

### 3.1 Grid-point model Meso-NH

To perform the grid-point simulation of the ABL, we have used the research model Meso-NH. It is a mesoscale atmospheric model jointly developed by the Laboratoire d'Aérologie and by CNRM-GAME (Lafore et al., 1998). It incorporates a non-hydrostatic system of equations, and simulates the dynamics of atmospheric motion over scales ranging from large (synoptic) to small (large-eddy). Meso-NH has a complete set of physical parametrizations for sub-grid modeling. The Meso-NH model is thus a reference tool for

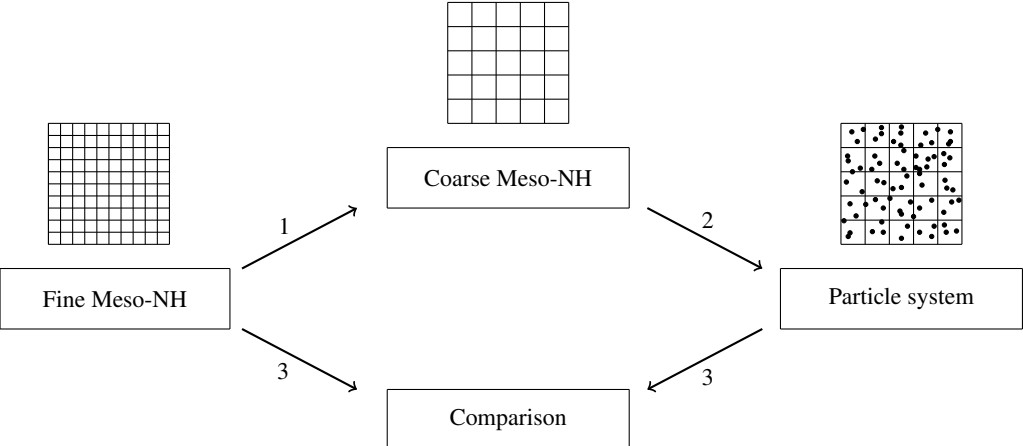

**Figure 2.** Our study framework. We use a high-resolution model, which is averaged to obtain a coarse field. Particles are forced with coarse fields and then fine fields are compared to particle fields.

modeling turbulence. It has already been used to evaluate the quality of the turbulence representation in models at kilometer scales (Honnert et al., 2011). Furthermore, Meso-NH can be used in a LES mode (Couvreux et al., 2005). Then, its high-effective resolution enables modeling of the main turbulent processes.

The equation system solved using the Meso-NH model is an approximated form of the Durran (1989) version of the anelastic system. Meso-NH is an Eulerian model which uses a fourth-order centered advection scheme for the momentum components and a finite-volume method for advection of meteorological variables (e.g. temperature, water species, turbulent kinetic energy) and passive scalars (Colella and Woodward, 1984). In order to suppress the very short wavelength modes, the model uses a fourth-order diffusion scheme applied only to the fluctuations of the wind variables.

For our simulations, the 3D turbulence scheme is a one and a half order closure scheme (Cuxart et al., 2000). Thus, the sub-grid TKE is a prognostic variable whereas the mixing length is a diagnostic variable. The mixing length and the dissipative length are computed separately according to Redelsperger et al. (2001). The mixing length is given by the mesh size depending on the model dimensionality. This length is limited to the ground distance and also by the Sommeria and Deardorff (1977) mixing length, which is pertinent in the stable cases. The eddy dissipation rate is computed from the sub-grid turbulent kinetic energy using a closure scheme based on the mixing length.

### 3.2   Model coupling

To introduce the presentation of the Meso-NH simulation which has been used, the framework of the coupling scheme is described.

Our scheme is a first step in coupling particle systems and grid-point models. Here, we work on the downscaling from the grid-point model to the particle system in one way, so the information flow goes only from the grid-point model to the particles. We have used simulations of the convective boundary layer to force a particle system which models sub-grid turbulent phenomena.

The particle systems are forced with a large-scale grid-point meteorological fields. The large scale used in our work is later described in details. The forced particles are used to model the sub-grid fields for the large-scale model. To validate the downscaling, a higher resolution model is used. In theory, the turbulent fields represented by the particles should be compared to the same fields simulated by a high-resolution grid-point simulation. For computational reasons we do not have access to different high-resolution simulations. Thus, the large-scale simulations have been built from the available simulations.

The process consists of three steps as outlined in figure 2. First, we have performed high-resolution simulations with the Meso-NH model. These simulations are the finest available simulations. Hence, they are considered as a reference and represent the real atmosphere. The high-resolution simulations are not directly used to force the particles. Thus, they can be used to independently assess the turbulence modeled by the particle system.

Now that we have a reference simulation, a coarser simulation is built in order to force the particle system. To this end, we have chosen to average the grid-point fields of several cells. To be consistent, we have also applied a temporal average. The resulting coarse Meso-NH fields have thus lower spatial and temporal resolutions than the reference simulation while being consistent with it. However, due to the average, the coarse fields include not only the components resolved on the grid, but also the average of the sub-grid components. This limitation will be discussed in section 8.

In each cell of the coarse grid and during each coarse time step, the particle average behavior is forced. If the forcing

method works, the average of the particles should be in good agreement with the coarse fields.

To assess the method, we have worked on wind fields and turbulent kinetic energy (TKE) fields. As explained in section 4.2, the TKE and the horizontal wind are not directly forced by the coarse model. The models and the data we used are presented in the two following sections.

### 3.3 The particle system

In order to model the sub-grid processes, an applied mathematical technique is used : the probability density functions are described using a particle system. In this study, the particles sample the wind probability density function. The particle technique is widespread in research fields such as mechanical system modeling (automotive, aeronautics), but it is not yet currently used for atmospheric modeling. However, Lagrangian particle models for dispersion have been discussed for quite a long time. Guidelines for evaluating the relevance of a stochastic model have been given by Thomson (1987), and different particle pair models for dispersion and concentration fluctuations are described in Thomson (1990) and Durbin (1980). These models have then been improved and generalized to particle system models for dispersion and air pollution modeling (Uliasz, 1994; Stohl and Thomson, 1999).

Here, a particle is a realization of the surrounding atmosphere. Depending on the complexity of the evolution model, the particles carry physical properties, such as fluid velocity, temperature or humidity rate. In our study, the evolution model is simple, and each particle is a position/velocity couple. Using the particle approximation of the probability density function and the physical properties of the particles, the statistics of the turbulence are computed. In particular, the wind variance can be computed. Thus, using the particle wind, the TKE is directly available.

The suggested stochastic downscaling method completes the long list of downscaling techniques developed to improve geophysical model resolutions. Among them, we find the adaptive mesh refinement for oceanic and atmospheric models (Blayo and Debreu, 1999; Debreu et al., 2005; Andrews, 2012). Our downscaling method offers an other point of view. Instead of refining the grid, the sub-grid atmosphere is modeled using a particle system which lives inside the grid cells.

Previous studies have already explored stochastic downscaling methods for the meteorological model MM5 (see Rousseau et al. (2007), Bernardin et al. (2009), and Bernardin et al. (2010)). Like these studies, our work aims at modeling the wind at very small scales on a limited area. The same particle evolution model is used. In these studies, the downscaling is performed by imposing the boundary conditions. The conditions are given by a large-scale Eulerian model. It ensures the consistency of the particle system with the large-scale model. Moreover, it keeps particles inside the

simulation domain. Indeed, the forcing induces a reflection of the particles against boundary.

Different choices have been made to develop the presented downscaling method. We have chosen to force the particles cell by cell using the grid-point fields, but there is no imposed condition to the edge of the domain. The particles live freely in the domain and may go from one cell to another. When some particles go out the domain, they are deleted and replaced by new particles inside the domain. For each new particle, the particle position is randomly chosen. Then, the new particle velocity is computed using the velocities of the particles which are in the same cell. Thus, the particle system contains information relative to different scales, including local components of the fields associated to sub-grid scale and large or mean components coming from the forcing, associated to the grid scale.

To compare the fields represented by the particles to the grid-point fields modeled with Meso-NH, average particle velocities are computed cell by cell. For instance, the wind $\mathbf{V}_\alpha$ represented by the particles in the cell $\alpha$ is given by the following expectation :

$$\mathbf{V}_\alpha = \frac{1}{N} \sum_{i=1}^{N} \mathbf{V}_\alpha^i$$

where $N$ is the number of particles in the cell $\alpha$, and $\mathbf{V}_\alpha^i$ is the velocity of the particle $i$. If the sub-grid processes are rightly modeled by the particle system, the fields represented by the particles should be similar to the fine Meso-NH fields. The results of the downscaling are presented section 6. They are obtained with 75 particles in each fine-grid cell. Thus, the whole system contains 19200 particles. This number may be compared to the 800 particles per grid cell used by Bernardin et al. (2009) for the same kind of application.

### 3.4 The Stochastic Lagrangian Model

For the particles to be realizations of the surrounding atmosphere, the particle evolution is driven by a local turbulence model. It is a Stochastic Lagrangian Model (SLM) inspired from Pope (2000) and introduced for atmospheric turbulence estimation purpose by Baehr (2009, 2010). The model is consistent with Kolmogorov's similarity hypothesis for turbulence (Kolmogorov, 1941). It describes the evolution of the position, $X$, and the 3D velocity of each particle. The position evolution is done by integrating the velocity. The velocity is split in one term for the horizontal velocity, $V$, and an other for the vertical velocity, $W$. Their evolutions depend on the local properties of the atmosphere and also on the atmospheric large-scale characteristics.

In the evolution equations, the large-scale influence is given by the pressure gradient $\nabla_x \overline{p}$ for the horizontal velocity, and by the mean vertical velocity increment $\Delta_k W$ for the vertical velocity at the time index $k$. The velocity evolution depends on the local properties through a term of wind fluctu-

ation around the locally averaged wind, $< V >$ and $< W >$. The local average operator $< . >$ is described in the next section. Then, for the vertical velocity, the buoyancy effect is taken into account. The last term of the equations is a dispersion term. It is represented by a Wiener process $\Delta B^\bullet$ normalized by the time step $\delta t$. Finally, the SLM equations are given by :

$$\begin{cases} X_{k+1} & = \quad X_k + \mathbf{V}_k\,\delta t + \sigma^X \Delta B^X_{k+1} \\ V_{k+1} & = \quad V_k - \nabla_x\overline{p}\,\delta t \,-\, C_1\frac{\varepsilon_k}{K_k}\,[V_k - <V>]\,\delta t \\ & \quad + \sqrt{C_0.\varepsilon_k}\,\Delta B^V_{k+1} \\ W_{k+1} & = \quad W_k + \Delta_k W \,-\, C_1\frac{\varepsilon_k}{K_k}\,[W_k - <W>]\,\delta t \\ & \quad + \frac{g}{\beta}\Gamma^\theta_k\,\delta t + \sqrt{C_0.\varepsilon_k}\,\Delta B^W_{k+1} \end{cases}$$

where $g$ is the standard gravity, $\varepsilon_k$ is the eddy dissipation rate (EDR) and $K_k$ the TKE. $C_0$ is the Kolmogorov constant. The constant $C_1$ is given by $C_1 = \frac{1}{2} + \frac{3}{4}C_0$ as suggested by Pope (1994).

The buoyancy effect term $\frac{g}{\beta}\Gamma^\theta_k$ is not directly modeled here. It is replaced by a random variable. We have chosen to use a centered Gaussian variable. This choice has been made for the sake of simplicity. However, an equation for the temperature evolution may be added to improve the buoyancy-effect modeling. Note the noise term at the end of the position evolution equation, which has been included to take into account the velocity integration errors considering the Euler scheme used for the velocity equation (see Bally and Talay (1996)).

The stochastic Lagrangian model has two control parameters for the velocity equation. The EDR, $\varepsilon$, and pressure gradient, $\nabla_x\overline{p}$, are the control parameters for the horizontal velocity. For the vertical velocity, they are the EDR $\varepsilon$ and the velocity increment $\Delta_k W$. In our downscaling method, they are given by the Meso-NH coarse simulation : it is how the coarse model forces the particle system. The two control parameters are related to the different scales modeled in the Meso-NH simulation. The pressure gradient and the vertical velocity increment are related to large scales, whereas the EDR is related to small scales. Thus, each parameter is associated to an extremity of the energy cascade described by Kolmogorov. The pressure gradient has be chosen to be consistent with the model described by Pope (2000), but the mean horizontal velocity can be used instead in case of difficulty in computing the pressure gradient. We also underline that the EDR used to force the system is a diagnostic variable of Meso-NH. Therefore, it is computed using a closure scheme. This choice is discussed in section 8. Contrary to the EDR, the TKE, noted $K$ in the equations, does not come from the Meso-NH simulation. Instead we use the TKE computed using the particle system and the local average operator described in the next section.

### 3.5 Ensemble averaging

As we have seen, the SLM equations contain some locally averaged terms, denoted by $< \cdot >$. In our framework, the fields are represented by the particle system. Thus, only discrete representations on irregularly spaced points of the fields are available. It leads to a tricky implementation of the average. A regularization function $G^\delta$ is introduced. To compute the average at a point $x$, this function gives a weight to the particles depending on their distance to $x$. Then, the local average is the weighted average of the particle values. The regularization function $G^\delta$ is a Gaussian which is centered at the computation point $x$. The variance of the function is noted $\delta^2$. The standard deviation $\delta$ is a length which depends on the homogeneity of the medium. Therefore, if the medium is homogeneous, the average can be computed using all the particles. In this simple case, the length $\delta$ can be long. On the contrary, if the medium presents strong spatial variations, only very close points have to be taken in account to get a representative average. Then, the length $\delta$ has to be short.

To validate the downscaling method, the particle fields are compared to the fine Meso-NH simulation. In a grid-point model, the characteristic length of the modeled processes is twice larger than the grid size according to the Nyquist's frequency (Nyquist, 1924, 1928). Thus, the grid size may be seen as the characteristic length, and we set the length $\delta$ at the finest Meso-NH grid size. As the horizontal grid and the vertical grid have different sizes, the horizontal characteristic length is set at $\delta_h = 40$m and the vertical one is set at $\delta_v = 12$m.

The average operator $< \cdot >$ is used to compute any needed structure functions. In particular the velocity variance may be computed using the local average and the particle system. The TKE is thus available at each time step. The ensemble averaging computation is independent of the forcing model grid. The particle system is viewed as a whole, and the local average may be computed using particles which are in different cells.

To compare the particle TKE to the TKE simulated by Meso-NH, the particle values are averaged cell by cell as explained in section 3.3.

## 4 The forcing

### 4.1 Meso-NH simulation

As Meso-NH is a research model, its grid size and its time step may differ from one simulation to an other. For the simulation used here, the horizontal grid size is $\Delta_x = 40$m, and the vertical one is $\Delta_z = 12$m. This simulation uses the set up of Darbieu et al. (2015), and uses a 256x256x256 points domain with cyclic conditions. The simulation starts at 06h00 and lasts 15 hours. Because of the high computational time of the simulations, we present only 15 minutes of simulation with a $\delta_t = 5$s time step. Moreover, to avoid the explosion of the computational time of the forcing procedure, we have only kept a 8x8x4 grid points. The first vertical level altitude is around 360 meters. The data are chosen in the middle of

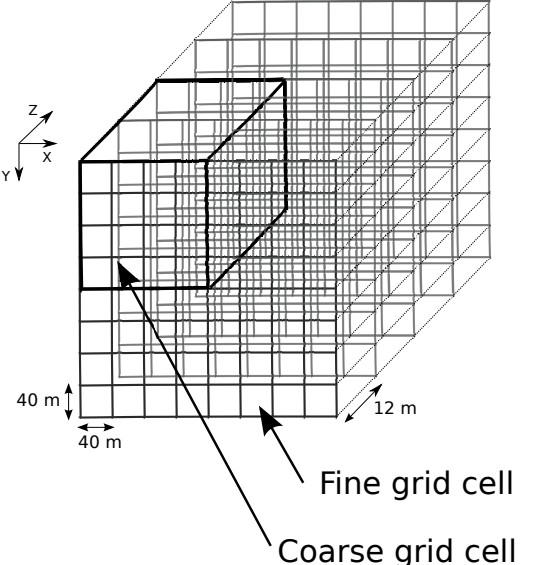

**Figure 3.** The domain of simulation : 8x8x4 fine cells which are grouped in 2x2x2 coarse cells.

the ABL for two reasons. First, at these heights, issues linked to the vicinity of the ground are avoided. In addition, in convective conditions, the turbulence is well established in the middle of the ABL. In the following sections, this configuration of Meso-NH is called the fine reference configuration. It will be used to evaluate the fields reconstructed with the particle system.

To force the particles, averaged fields are used. They are deduced from the reference fields. The averaged fields, called coarse fields, are obtained by averaging several grid points, on 12 time steps ($\Delta_t$=1min). Each average is made on a 4x4x2 grid-points domain (figure 3). So, the coarse grid size is 160mx160mx24m.

We have selected data from 13h55 to 14h10 during the convective period, the 06/20/2011. Among all the available variables in Meso-NH, we have extracted the atmospheric pressure, the 3 components of the wind, the TKE and the eddy dissipation rate (EDR), for the reference configuration. The pressure gradient is computed using the pressure field. The previously described average has been applied to these reference fields to obtain the coarse fields. We underline that only the coarse fields are used to force the particle system. The reference fields are simply used to evaluate the fields obtained from the particles.

To ensure its consistency with the real case, the high-resolution Meso-NH simulation has been compared to another LES simulation performed with the LES model of the National Center for Atmospheric Research (NCAR) for the same case and shows similar results (Darbieu et al., 2012)

### 4.2 The particle system forcing

The aim of this work is to study the ability of a particle system forced by grid-point data to model the sub-grid processes. To do so, we use two grid-point simulations : a coarse one to force the particle system and a fine one to assess the fields reconstructed by the particles.

The starting point of our downscaling scheme requires having the large Meso-NH simulations on a 3D domain including $N_c$ coarse cells. The time step of these simulations is denoted $\Delta t$. The particles evolve freely in the $N_c$ cells with a time step $\delta t$, shorter than $\Delta t$. The downscaling method used in this paper involves the following four steps :

1. initializing the particles in each cell using velocities given by the coarse Meso-NH simulations,

2. performing particle evolution with the SLM model and the time step $\delta t$,

3. calculating the sub-grid wind and the sub-grid turbulent parameters using the local average operator,

4. updating the values of $\varepsilon$, $\nabla_x \overline{p}$ and $\Delta W$ when the time $\Delta t$ is reached otherwise going back to step 2.

At the scale of the particles, the coarse grid-point data represent an averaged forcing. Note that the particle horizontal velocities are not directly forced with coarse winds. The horizontal velocities are forced with the pressure gradient and the dissipation rate. For the vertical velocities, the forcing is slightly different : it uses the vertical velocity coarse fields instead of the pressure fields. This choice has been made because horizontal velocities are driven by pressure gradients, whereas vertical velocities are driven by the buoyancy. To improve the downscaling method, temperature gradients computed from Meso-NH simulations could be taken into account. Note that in this work, the EDR used to force the particles is considered isotropic.

During 12 time steps, the values of the control parameters remain constant. To compute the particle simulation, steps 2 and 3 are repeated in a continuous loop until the time $\Delta t$ is reached. Then, the control parameter values are updated before computing the next 12 time steps $\delta t$ (figure 4).

In this procedure, the particle management is hidden. In our simulations, the particle number is constant. In practice, we have to ensure that all the particles are in the simulation domain. In our work, the particles follow the simulated airflow. So, at each time step some particles leave the domain. The outside particles are replaced by new particles with consistent positions and velocities as explained in section 3.3.

As the particles evolve freely in the domain, we also have to ensure a homogeneous repartition of the particles inside the domain. To do so, for each cell of the fine grid we keep the particle number between a minimal value and a maximal value which are given at the beginning of the simulation. By

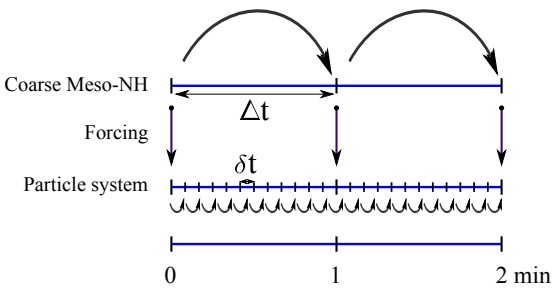

**Figure 4.** The different time scales used in this work. The forcing is applied at each coarse time step $\Delta t$. Between two coarse time steps, particles evolution is performed with a fine time step $\delta t$ and sub-grid processes are modeled.

displacing particles, this method of particle management limits trajectory length and prevents rogue trajectories described by Yee and Wilson (2007),Postma et al. (2012), and Wilson (2013).

## 5 Turbulent kinetic energy

In this section we first review how TKE is computed in Meso-NH. Then, we present the TKE computation using the particle system.

### 5.1 The TKE in Meso-NH

To characterize turbulence, the TKE and the EDR are the two parameters usually used. The TKE is the turbulent kinetic energy associated with the small-scale turbulent structures, while the EDR quantifies the energy transfer from the large-scale structures to the small-scale structures.

In this work, the two turbulent parameters play different roles. The EDR is used to force the particle system, whereas the TKE is used to assess the particle representation. For the EDR fields, we use directly the Meso-NH variable. It is computed from the TKE using a mixing-length closure hypothesis. We give now details about how the TKE is computed.

The TKE modeled by Meso-NH is made of two terms : the resolved TKE and the sub-grid TKE. The resolved TKE is a diagnostic variable. It is calculated using the grid-point 3D wind field $(u, v, w)$ :

$$K_{res} = \frac{1}{2}(\overline{(u - \overline{u})^2} + \overline{(v - \overline{v})^2} + \overline{(w - \overline{w})^2})$$

where the bar indicates a spatial average. Here, as the domain of simulation is very small, we choose to compute the average on all the 8x8x4 grid points.

The sub-grid TKE, $e$, is a prognostic variable of Meso-NH which is computed using a parametrization (Cuxart et al., 2000). In Meso-NH, the total TKE for the cell $\alpha$ is given by the sum of the sub-grid TKE and the resolved TKE :

$$K_\alpha = K_{res} + e_\alpha$$

where $e^\alpha$ is the sub-grid TKE for the cell $\alpha$.

In the Meso-NH simulations, the grid size is fine and the resolved TKE is the major contribution to the total TKE, as expected far from the surface layer.

### 5.2 The TKE modeled by the particle system

The particle system is used here to model the wind inside the grid Meso-NH model. As detailed in section 3.5, using the wind modeled by the particles, the total TKE is directly available. The TKE associated with particle $i$ is computed at each time step as follows :

$$K^i = \frac{1}{2} < (u^i - < u >)^2 + (v^i - < v >)^2 + (w^i - < w >)^2 >$$

where $< . >$ represents the local average. For the cell $\alpha$, the TKE is thus given by :

$$K_\alpha^i = \frac{1}{N} \sum_{i=1}^{N} K^i$$

where $N$ is the particle number in the cell $\alpha$. Therefore, the TKE computation may be adapted to the grid size, and the particle fields may be compared to coarse and fine fields. Thus, by construction, the TKE modeled by the particles contains small-scale contributions –sub-grid for Meso-NH– and large-scale contribution –resolved by Meso-NH.

## 6 Downscaling results

In the previous sections, the downscaling algorithm has been described in details. In this section, the downscaling results are presented. To assess the behavior of the particle system, we compare the 3D wind given by the particle and by Meso-NH. First, results on the coarse grid are shown. Then, we present the comparison between the particle fields and the fine Meso-NH fields. Wind power spectral densities are then presented. Finally, results for sub-grid TKE are presented. These results are presented separately because the sub-grid TKE is computed from the particle TKE and not directly from the wind.

### 6.1 3D wind results

To illustrate the results of the downscaling scheme, the wind results on the coarse grid and on the fine grid are presented for one cell and for the 3-dimensions.

#### 6.1.1 Coarse grid

To model the sub-grid fields, the particles are forced by coarse grid-point fields of pressure gradient, EDR, and mean vertical velocity increment.

To assess the downscaling method, the first thing to look at is the agreement between the coarse wind and the average

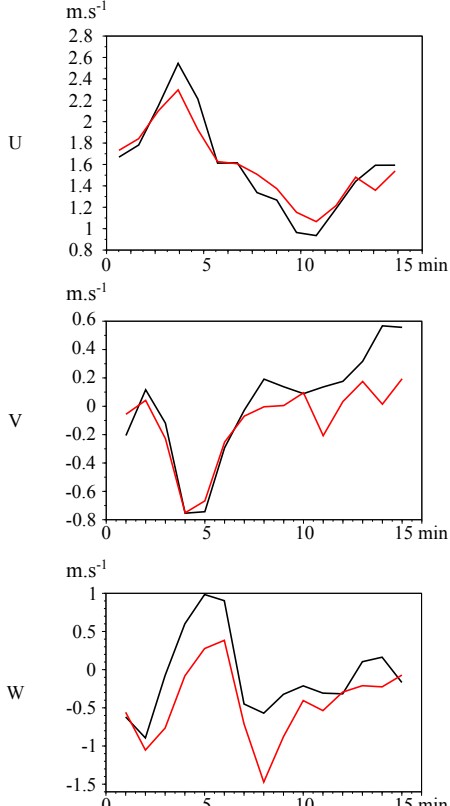

**Figure 5.** Time evolution of the three wind components obtained from the coarse Meso-NH simulation (black) and by the particle model (red) for one cell of the coarse grid.

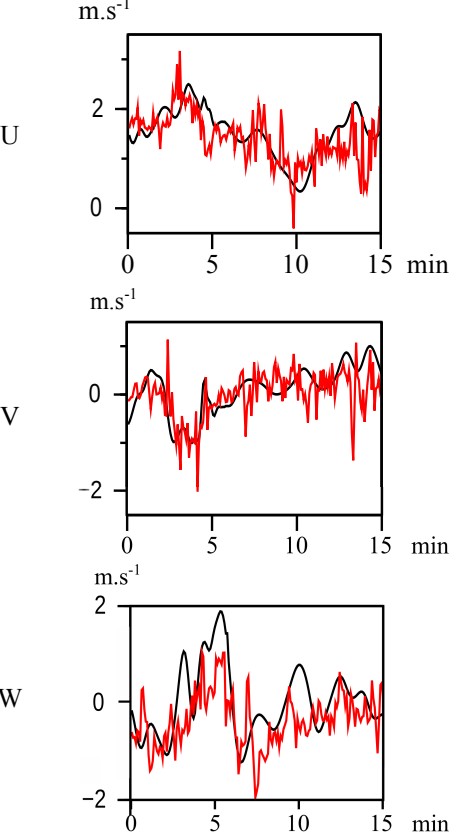

**Figure 6.** Time evolution of the three wind components obtained from the fine Meso-NH simulation (black) and by the particle model (red) for one cell of the fine grid.

of the sub-grid wind modeled by the particle system. The aim is to assess the particle behaviour at the forcing scale. This verification is important, especially for the horizontal velocity which is not directly forced by the coarse horizontal velocity fields. Recall that to compare the particle wind to the coarse wind fields, the particle values are averaged coarse cell by coarse cell.

Figure 5 compares the three components of the wind modeled by the particles and by Meso-NH on one cell of the coarse grid. The averaged particle wind is consistent with the coarse wind, especially for the horizontal wind. The root mean square errors associated to the first and the second particle wind components are respectively $0.045 m.s^{-1}$ and $0.062 m.s^{-1}$. For the vertical wind, there is more discrepancy between the particle wind and the Meso-NH wind, but they present the same variations. The associated error is $0.135 m.s^{-1}$. Similar results have been obtained for the other coarse cells (not shown). Thus, as expected, the 3D wind modeled by the particles is in good agreement with the wind modeled by the coarse Meso-NH model.

### 6.1.2 Fine grid

We are now interested in the particle behaviors at the fine scale. Here, the particle values are average fine cell by fine cell to obtain wind fields at the fine Meso-NH resolution. The particles are forced using only coarse Meso-NH fields. Thus, the particle fields could differ from the reference fine Meso-NH fields.

In figure 6, the three wind components are represented for one cell of the fine grid. First, we notice the more turbulent profile of the 3D wind represented by the particles than the fine Meso-NH wind profile. Indeed, the Meso-NH wind appears smoother while the particle wind presents more temporal fluctuations. The interpretation of the power spectral densities presented section 6.1.3 confirms that the energy associated with high frequencies is higher in the particle wind than in the reference Meso-NH wind.

The reference fields are represented with a 5-second time step. At this frequency, it appears that the Meso-NH wind is smoother than wind usually observed in the boundary layer. By comparison, the fluctuating profile of the particle wind

seems consistent with wind observations obtained by a 3D sonic anemometer mounted below a tethered balloon (see Canut et al. (2016) for instance).

To explain the smoothness of the Meso-NH wind, two comparisons have been done. First, we have compared the wind fields from the studied area to fields from different areas at the same vertical level. Subsiding or ascending areas have been chosen. It appears that Meso-NH produces a smooth wind, in both ascending and subsiding areas. Next, a simulation without numerical diffusion was performed, and the modeled winds with and without diffusion have been compared. This comparison shows that the smoothness is not due to the numerical diffusion used in the simulations.

Coming back to figure 6, we can see that the particle wind seems to follow the fluctuations of the Meso-NH wind. This remark leads us to look at the low frequency component of the particle wind. In section 6.1.3, the difference between the particle wind and its low frequency component is also investigated.

To check that the particle wind follows mainly the Meso-NH wind fluctuations, we apply a low pass filter on the particle wind. The aim is to suppress the fast fluctuations and then to assess the low-frequency component of the particle wind. A second order low pass filter with a cutoff frequency of $2.10^{-2}s^{-1}$ has been used.

The results are presented in figure 7. Thanks to the low pass filter we are able to easily compare the particle wind time series to the Meso-NH wind time series. The low-frequency component of the particle wind presents the same variations than the reference Meso-NH wind. However, one can notice that the results are better for the horizontal wind components than for the vertical. One way to improve the vertical velocity agreement is supplement the SLM with a model that includes buoyancy effects. According to these results, the filtered particle wind seems consistent with the fine Meso-NH wind.

The particle wind fields do contain the same low frequency information than the fine Meso-NH wind fields. Thus, the suggested downscaling method and the model coupling have worked. Comparing to the Meso-NH wind, the particle wind has a faster fluctuating component. The question is now to determine if the fast fluctuations are due to smaller turbulent structures than those modeled by Meso-NH or if they are only a noise added to the low frequency signal.

To propose a beginning of an answer, we present the study of the power spectral densities (PSDs) of the wind and its low and high-frequency components in section 6.1.3.

### 6.1.3   Validation using PSD and wind anomalies

To further the comparison between the Meso-NH wind and the particle wind, we have computed the wind PSDs. First, spectra of time series have been studied. Then, the PSD of the wind anomalies – differences between the particle wind and its low frequency component– are shown. Finally,

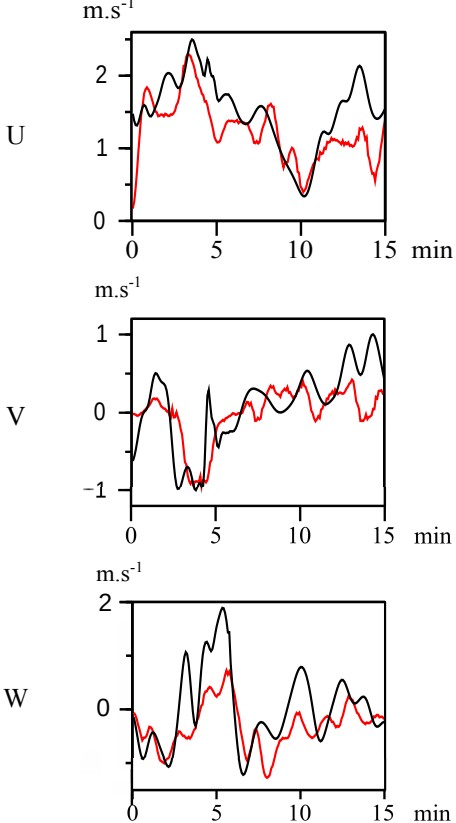

**Figure 7.** Components of the fine Meso-NH wind in black and components of the particle wind in red, for one cell on the second level of the fine grid.

we discuss briefly the effective resolution of the Meso-NH model by comparison with the LES model of the NCAR.

To assess the temporal variability of the particle wind, time PSDs are computed. For each vertical level, the PSDs are computed using groups of 4x4 fine grid cells. Each of these groups contains fine cells which are forced by a same coarse cell. A Fourier transform has been applied to time series of each fine cell. Then, the Fourier coefficients of the 4x4 cells are averaged. This operation gives 4 PSDs per vertical level. The aim is to check the consistency of the different winds with the K41 theory (Kolmogorov, 1941).

Figure 8 presents the PSDs of the three components of the fine Meso-NH wind and of the particle wind. It appears that none of the particle-wind or the fine Meso-NH wind follows perfectly the energy cascade given by the Kolomogorov's theory and represented by the -5/3 slope. But, we may notice that the particle wind spectra present a regular slope. The regularity of the PSD slope shows that the energy cascades are the same whatever the frequency. Therefore, the regularity of the PSD slope is a good way to assess the quality

of the wind retrieved by the particles. However, the slope is sightly more gentle than the energy cascade. There may be too much energy associated with the turbulence modeled by the particles at high frequencies. As presented in section 3, the SLM has been designed to follow Kolmogorov theory, but the spectra in figure 8 have been obtained by applying the model in a new framework. Our work is a first attempt to use control parameters given by an Eulerian grid point simulation. In this framework, the model behaviour has not been completely assessed yet. Longer simulations are needed to continue working on it.

Contrary to the particle wind, the Meso-NH wind spectra have a "spoon shape" : they follow the -5/3 slope at low frequencies, then the spectrum slopes become steeper and finally the slopes are almost horizontal at high frequencies. Thus, the PSDs show that Meso-NH correctly represents the low-frequency components of the wind but the modeling worsens gradually following the frequency increase. At high frequency, the Meso-NH wind PDSs look like low energetic white noises. From the spectral analysis, we may deduced an effective temporal resolution of Meso-NH is about 50 seconds or $10.\delta t$. Hence Meso-NH does not model the high-frequency components of the wind correctly. Therefore, it may explain the difference between the particle wind and the Meso-NH wind we have seen in the previous section.

The spectral analysis of the time series has clarified the validity domain –in terms of temporal resolution– of the two simulations. It also shows the limit of the grid-point model for high frequency wind fluctuation modeling.

The spatial resolution of the particle simulations is trickier to estimate. A first estimation may be given by the Lagrangian lengths associated with the wind components. The lengths can be evaluated using the power spectral densities and the mean velocities. Looking at the spectrum of the first component of the wind, we can see that the spectrum is flat for frequencies higher than $5.10^{-2}\mathrm{s}^{-1}$ (figure 8). On average, over the domain, the first component of the particle wind is about $1.3m.s^{-1}$. Thus, a Lagrangian length associated to the particles for the first wind component is about 26m. Using the same cut-off frequency, we obtain Lagrangian lengths about 6m and 4m for the second and the third wind component respectively. The difference between the lengths computed for the three components clearly shows the difficulty in using this method to evaluate the spatial resolution of the particle simulations.

To get an idea of the Meso-NH's effective spatial resolution, we have compared the Meso-NH simulations to other simulations performed for the BLLAST experiment. Spatial PSDs are computed for a given time step on the whole domain –256x256x256 cells– using rows or columns of the fine grid. As for the time PSD, spatial PSD are performed using averages of Fourier coefficients. The PSDs along the x-axis are based on the averages of the Fourier transforms of the

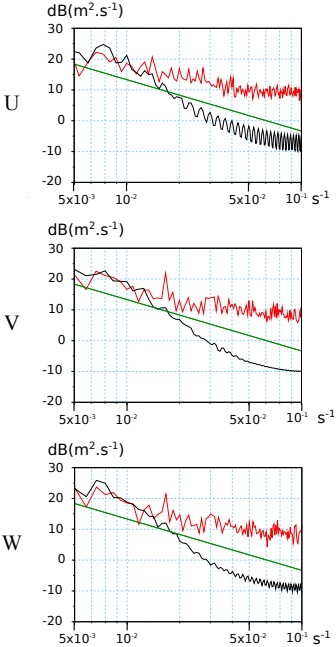

**Figure 8.** Power spectral densities of the components of the fine Meso-NH wind in black and of the components of the particle wind in red, calculated on a 4x4 grid cells domain, for the second level of the fine grid. In green we have the -5/3 slope according to the K41 theory.

rows. Respectively, for the PSDs along the y-axis we use the averages of the Fourier transforms of the columns.

In this paragraph, the particle wind PSDs are not available. Indeed, the downscaling scheme has been applied to a restricted domain which is far too small to compute the spatial PSDs.

Thus, to have a comparative element, simulations of the NCAR LES are used (Moeng, 1984). These simulations were performed by Darbieu et al. (2015). The spatial PSDs are computed for the two models using data from 06/20/2011 at 14h00 for the vertical level around 360 meters. Figure 9 shows the PSDs obtained for the three components of the wind following the two horizontal directions. The Meso-NH spectra and the NCAR LES spectra perfectly follow the expected energy cascade at low and medium frequencies.

At high frequencies, their shapes differ. From its formulation, the NCAR LES model spectra show a clear cut-off frequency. This frequency is around $8.10^{-3}m^{-1}$. The Meso-NH spectra show instead gradual decreases, but the spatial resolution seems almost equivalent to the NCAR LES model. According to the NCAR LES cut-off frequency, the effective spatial frequency of the Meso-NH model is about 125m. Thus, it leads to an effective resolution of $3.\Delta x$, which is in good agreement with previous studies (Ricard et al., 2013).

We may also notice the asymmetry of the spectra. It shows that the structures in the boundary layer are organized

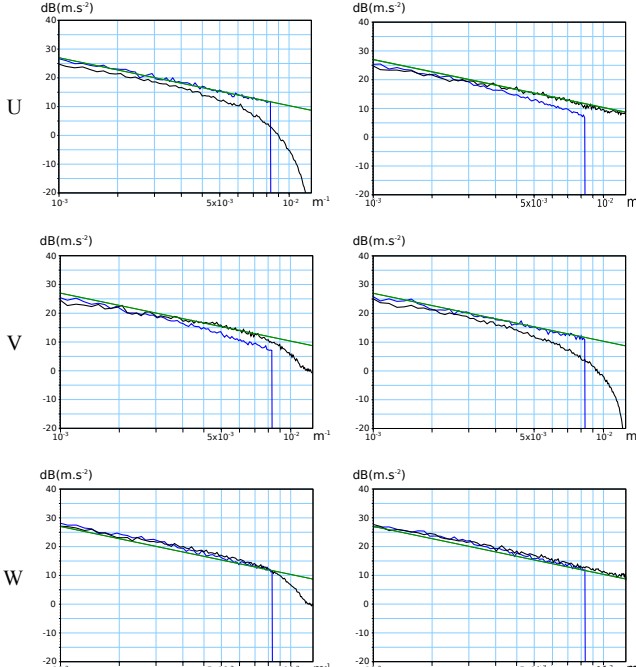

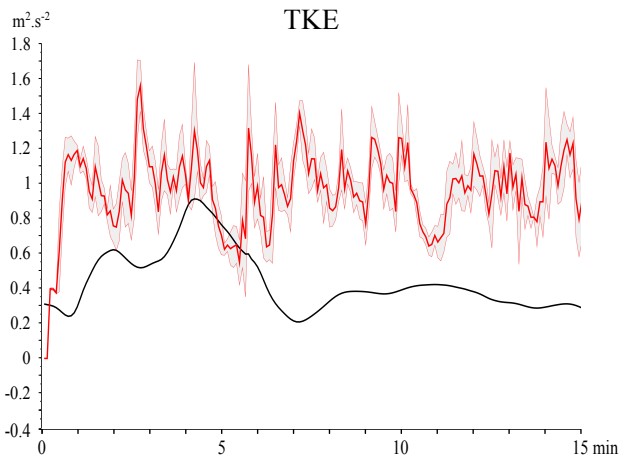

**Figure 10.** Fine Meso-NH TKE in black and particle TKE in red, for one cell of the fine grid. The gray zone represents the standard deviation of the particle TKE, for the particles in the cell.

**Figure 9.** "Row-average" (left) and "Column-average" (right) spatial PSD on the 256x256 cells grid of the components of the wind for Meso-NH in black and the LES model of the NCAR in blue. A green line indicates the -5/3 slope according to the K41 theory.

following preferential directions.

The study of the spatial spectra has shown that Meso-NH is able to model the spatial variability of the wind with a $3.\Delta x$ resolution. However, Meso-NH is not able to model the local wind fluctuations under its effective resolution. Once again, it explains why the Meso-NH wind simulation is smoother than the particle simulation.

To end the validation of the particle wind, we have studied the wind anomaly PSDs. The wind anomalies are defined here as the difference between the wind and its low frequency component. The study shows that the time PSDs of the anomalies follow the energy cascade. So, the anomalies are not a white noise. Thus, the particles do not add a simple noise to the coarse wind. The added information is in good agreement with the Kolmogorov K41 theory. It illustrates the effectiveness of the suggested downscaling method.

### 6.2   Turbulent kinetic energy results

In this section, the TKE simulated using the particle system is presented. As explained in section 4.2, the TKE is not directly forced by the coarse Meso-NH model. The particle TKE is computed at each fine time step $\delta t$, as it has been described in section 5. Then, the particle TKE is compared to the fine Meso-NH TKE.

The results are presented in figure 10. In this figure, we focus on the same cell as shown in the previous sections. First, Note that the particle TKE is double the Meso-NH TKE. The particle system technique also models an important time variability of the TKE, whereas the Meso-NH TKE is rather smooth. The particle system models more small-scale turbulence than Meso-NH. Thus, the results for the TKE modeling are in good agreement with our previous remarks on the wind results and the Meso-NH effective time resolution.

In Lothon et al. (2014) and Canut et al. (2016), studies of TKE evolution during the BLLAST experiment are presented. The authors give several observations time series of TKE. Their studies of TKE times series indicate that afternoon TKE is around $1 m^2.s^{-2}$, before decreasing at the end of day.

To force the particle system, we have used data from 360m and 400m high, from 13h55 UTC to 14h10 UTC. There is no TKE observation at this precise height during this period, but sonic anemometer and tethered balloon observations are available at several heights from 30m to 550m depending on the time. The TKE observations obtained using these instruments for the 06/20/2011 are given in (Canut et al., 2016).

Comparing the particle TKE to the TKE observations, we note that the particle TKE has the same order of magnitude as the TKE observed during the afternoon, from 0.6 $m^2.s^{-2}$ to 1.7 $m^2.s^{-2}$. A look at the Meso-NH TKE shows that the Meso-NH model seems to underestimate the TKE. This underestimation has already been described in previous works (Darbieu et al., 2015).

The comparison of the particle TKE with the observations shows encouraging results. These results are a first step to demonstrate the ability of the particle system to model very small-scale turbulence. However, to end the validation, the

suggested downscaling method will be applied to a larger domain and to other field experiment cases.

## 7   Sensitivity to the forcing grid

In the previous section, all the presented results have been obtained using the same forcing scale. Here, we suggest to briefly look at the influence of the forcing scale on the fields modeled with the particle system. As a first approach to qualify this influence, the particle system has been forced by two different scales.

The previously used grid was 160mx160mx24m large. The new grid used to evaluate the forcing scale influence is 80mx80mx24m large. This new grid is obtained by averaging the fine grid on 2x2x2 cells. The temporal resolution of the forcing is the same for the two experiences ($\Delta t$ =1min).

First, the particle winds are compared to the fine Meso-NH wind. The root mean square error (RMSE) between each particle wind and the Meso-NH wind are presented in table 1. The RMSEs of the low frequency components are also presented. To compute these RMSEs, we have only used the first 5 minutes of our downscaling simulations.

**Table 1.** Wind RMSE depending on the forcing scale.

|                               | 160m  | 80m   |
| ----------------------------- | ----- | ----- |
| RMSE – total signal           | 0.520 | 0.462 |
| RMSE – low-frequency signal   | 0.523 | 0.429 |

As expected, the particle wind obtained with the finest forcing grid is the closest to the Meso-NH wind. However, the difference between the two forcing methods are rather small. Using the finest grid reduces the RMSE of 12% for the total wind, and of 20% for the low frequency component.

The influence of the forcing scale on the TKE is illustrated in figure 11. The differences between the two forcing methods are mostly visible for the first minute of the simulation, which correspond to the first large time step. The TKE obtained with the finest forcing is closest to the forcing model TKE. This particle TKE is also less fluctuating than the TKE obtained with the largest scale forcing, but they reach the same order of magnitude.

According to these results, the two particle simulations are consistent. Reducing the forcing scale reduces the difference between the particle fields and the model fields. However, for the two forcing scales, the particle fields are more turbulent than the Meso-NH fields.

To complete the work on the sensitivity to the forcing scale, a forcing grid of 40mx40mx12m has also been used. The fields modeled with this forcing grid represent sub-grid fields for the fine Meso-NH simulation. As expected, the particle fields are more turbulent than the fine Meso-NH fields, but they are similar to the previous sub-grid simulations (not shown).

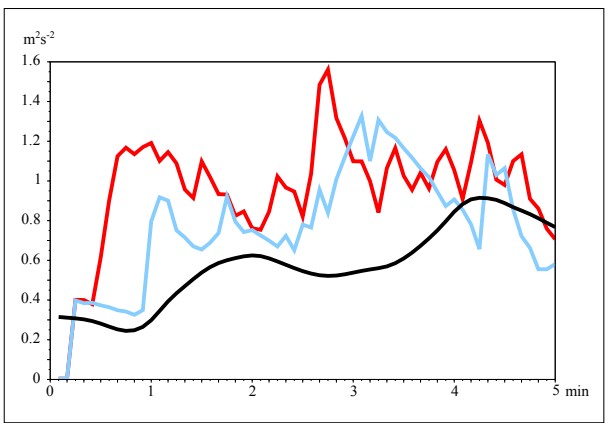

**Figure 11.** 5 minutes time series of TKE depending on the forcing scale : the TKE obtained with 160mx160mx24m is drawn in red, and the TKE obtained with 80mx80mx24m is drawn in blue. The fine Meso-NH TKE is drawn in black.

## 8   Discussion

This article presents a first work on a new way to model sub-grid processes using particle systems. One of the major improvements is the use of a simple turbulence model instead of complex model such as LES or DNS. However, to fully validate the method, one of the first steps should be to use a DNS or to apply the downscaling method to a toy model with known sub-grid fields. Unfortunately, such a validation has not been done yet.

For the work presented here, tests have been conducted on a small domain, with a reduced number of particles in each cell. These two constraints were related to computational time restrictions. Extending the domain and the duration of the simulation should be one of the next steps. It would improve the PSD quality, and limit the influence of the edges. Then, a supplementary work on the spatial resolution of the particle simulations should be done and , and the robustness of the results should be tested. To give a very first answer to the robustness issue, recall that we have compared the studied fields to fields from different areas at the same vertical level. The comparison has shown that Meso-NH fields are similar in the different areas. Thus, the downscaling method should provide similar results when being applied in these areas.

Related to the question of the spatial resolution of particle simulations, there is also the fundamental question of the scale of the turbulence represented in the particle fields. So far, only a first estimation of the scale has been given, and a specific work still needs to be done to figure out the scales represented by the particle model.

In this work, the coarse fields were computed by averaging the fine Meso-NH fields. In a more advanced exercise,

the coarse fields would be real Meso-NH fields computed with a coarse grid. To further this study, we could also add to the SLM an equation to model the temperature evolution. Therefore, the sub-grid buoyancy effect could be modeled and compared to a high-resolution Meso-NH simulation.

Concerning the SLM, another point has to be discussed. The Wiener processes used for the dispersion terms involve a locally-Gaussian assumption of the pdf described by the particles. In our work, the Gaussian assumption is not valid at the grid-cell scale. Indeed, at a given time in a given cell, particles with different characteristics are mixed. This is partially due to the free evolution of the particles in the domain. Thus, the velocity pdf described by the particles in one fine cell is obtained by mixing Gaussian pdf but it is not necessarily Gaussian.

We would like to underline an important point about the EDR fields used to force the particle system. Here, the chosen EDR is the Meso-NH variable. The advantage of this choice is that the EDR is directly available. However, as it is a diagnostic variable of the Meso-NH model, it is computed using a closure scheme. The closure scheme may induce errors in the EDR modeling due to the underlying assumptions. To control the assumptions which are made, we could compute the EDR from the grid point wind field, and compare it to the EDR calculated using different closure schemes. As the EDR controls the particle dispersion, an improvement in the EDR modeling will directly lead to an improvement in the sub-grid turbulence modeling.

Therefore, in future work, the downscaling method should be applied to a larger domain, and sub-grid fields should be compared to observations. In addition, an in-depth comparison of TKE parametrization used in Meso-NH and TKE modeled with the particle method should be conducted. Despite the computational time, from a long-term perspective future exercises should include replacing the sub-grid parametrization used in Meso-NH by sub-grid particle modeling. Indeed, for research purposes, the downscaling method may be an alternative solution to common turbulence closures which often assume isotropic and homogeneous turbulence.

## 9 Conclusions

We present here a new downscaling method based on the coupling of a grid-point model and a particle model. The downscaling method has been applied to a simulation performed for the BLLAST experiment.

The particle system has been forced by a coarse model. Then, the particle fields have been assessed against a high-resolution simulation. The particle winds seem in good agreement with the high-resolution winds, but higher resolution simulations should be performed. The same conclusions are given for the TKE simulations.

Even if the domain size is a limitation of the present study, the presented results are very encouraging. They prove the relevance of the suggested forcing method. Forcing a particle system is a quite simple process, and the sub-grid fields seem consistent with observations. Therefore, the first step to couple the SLM model and the Meso-NH model is achieved.

In a longer term, this work may be used to compare and to test the different turbulent schemes, parameterizations, or closure hypothesis available in the research models and in the operational weather forecast models.

*Acknowledgements.* BLLAST field experiment was made possible thanks to the contribution of several institutions and supports : INSU-CNRS (Institut National des Sciences de l'Univers, Centre national de la Recherche Scientifique, LEFE-IDAO program), Météo-France, Observatoire Midi-Pyrénées (University of Toulouse), EUFAR (EUropean Facility for Airborne Research) and COST ES0802 (European Cooperation in the field of Scientific and Technical). The field experiment would not have occurred without the contribution of all participating European and American research groups, which all have contributed in a significant amount. BLLAST field experiment was hosted by the instrumented site of Centre de Recherches Atmosphériques, Lannemezan, France (Observatoire Midi-Pyrénées, Laboratoire d'Aérologie).The 60m tower is partly supported by the POCTEFA/FLUXPYR European program. BLLAST data are managed by SEDOO, from Observatoire Midi-Pyrénées. The French ANR (Agence Nationale de la Recherche) supports BLLAST analysis in the 2013-2015 BLLAST_A project. The authors particularly thank Clara Dardieu for providing the NCAR LES simulations, and Bruno Piguet for the tethered balloon observations. The authors also thank the coeditor and the reviewers for their comments which have improved the quality of this article.

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
