# Peer review of "A new downscaling method for sub-grid turbulence modeling"

_Atmospheric Chemistry and Physics, 2015_

## Referee Comment (RC1) · Anonymous Referee #1 · 18 Feb 2016

**Review of acm-2015-1015**

**Overall Comments and Recommendation:**

In this paper, the authors have developed what they feel is a model for subgrid turbulence in a mesoscale atmospheric model. According to my understanding of the model, a coarse-grained turbulence model is first averaged, then used to drive a Lagrangian stochastic dispersion model which is said to represent the effects of subgrid scale turbulence. The authors note that this is a first step, and thus there is only one-way coupling, or in other words the Lagrangian particles are not used to send energy back to the larger scales. As viewed by the authors, the novel aspect of the paper is that, in contrast to previous work, no assumption is made about the shape of the probability distribution function for the subgrid scales.

Based on the description given of the modeling methods, I have some major concerns with the theory behind the work, which are addressed below. As such, I recommend major revisions before considering the manuscript for publication. I have also given some minor suggestions mostly regarding formatting that would be helpful if there is a future review. I have not provided an exhaustive list of minor revisions given the degree of revision suggested below.

**Major Comments:**

1. The methods described in this paper are not a "sub-grid turbulence model". This is simply a RANS Lagrangian particle dispersion model. Essentially, the authors average out the grid-scale turbulent motions, and instead use the mean fields to drive the Lagrangian dispersion model (at least, this seems to be how it is described in the text). Thus, the mean field is the 'resolved' component and the Lagrangian evolution equation is used to add in the 'fluctuating' component, or in other words the authors are using a Reynolds decomposition:

$$u_i = \langle u_i \rangle + u_i'. \tag{1}$$

where $\langle u_i \rangle$ is an ensemble average, and $u_i'$ is the fluctuation from that average. If I am understanding the paper correctly, the authors then prescribe $\langle u_i \rangle$ using the Eulerian simulations, and $u_i'$ is (indirectly) determined using the Lagrangian stochastic evolution equation.

In order for this to be considered a "subgrid scale" model, the following decomposition would be used

$$u_i = \tilde{u}_i + u_i'', \tag{2}$$

where now $\tilde{u}_i$ is the filtered component (filtered at the grid scale), and $u_i''$ is the unresolved or subfilter component, which would be determined by solving the evolution equation in terms of the subfilter component:

$$du_i = a_0 dt - a_1 \left(u_i - \tilde{u}_i\right) dt + b\, dW, \tag{3}$$

where $dW$ is an increment in a Weiner process, and coefficients $a_0$, $a_1$, and $b$ are determined such that Eq. 3 satisfies the Navier-Stokes Equations. For examples of determining the coefficients see Weil et al. (2004), Shotorban and Mashayek (2006), or Vinkovic et al. (2006). These examples are applied to traditional LES applications, but the principles are the same as in the present manuscript. Thus, the turbulence calculated at the grid-scale is retained, and modeling is reduced to specifying only the subgrid scale turbulence. Why filter out valuable resolved turbulence by averaging? The point of subgrid scale modeling is to retain as much information as possible, such that modeling is simplified in that we only have to model the smaller, more 'universal' scales.

2. The authors state that the novelty of the proposed modeling methodology is that (Lines 77-78) "The method we suggest differs from these previous works: no assumption is made on the pdf shape." This is not true. By using a Langevin-based equation for the particle motions that is forced by a Weiner process, *the authors are effectively specifying the shape of the pdf.* Although Sects. 2.2 and 2.4 of Pope (1994) (for example) states that these methods do not assume a pdf, the form used by the author does assume a pdf. Particularly, it is assumed that the pdf is Gaussian with variance of $\sigma^2 = \frac{C_0 K}{2C_1}$ (where of course $K$ is the turbulent kinetic energy). The authors can verify this for themselves. Take the Lagrangian evolution equation, simulate some particles, and calculate the pdf. I have attached some sample MATLAB code to do this in an appendix of this review. For simplicity I have assumed that $\varepsilon$ and $K$ are constant, and that there is no mean drift (i.e., $\nabla p = \langle U \rangle = 0$). Means could be included, but they should only shift the pdf and not affect its shape. We should find that an ensemble of particles should have a velocity pdf of

$$P = \left(2\pi\sigma^2\right)^{-1/2} \exp\left(-\frac{u^2}{2\sigma^2}\right), \tag{4}$$

where again $\sigma^2 = \frac{C_0 K}{2C_1}$ is the prescribed velocity variance. As a side note, this is not exactly true for the authors' equation for $W$ given the arbitrary way that they have included buoyancy (i.e., it is random and thus will generate erroneous additional variance). If it was correctly modeled, it should create a skewed velocity distribution and thus none of the analysis above or in the manuscript is valid. See Cassiani et al. (2015) and the references therein for examples.

The output of the MATLAB code verifies this (shown below). In essence, it seems that it's not necessary to simulate all the Lagrangian trajectories. Simply use the above equation to get the pdf. If the velocity is of interest, substitute $\sigma^2$ at any point (interpolated) and use the pdf along with a uniform random number generator to draw a random $u$.

[Figure]

3. In the (unnumbered) equations following line 348 (note that line numbers are messed up here, there are more than 5 lines between labels 335 and 340), why is an additional noise term added to the position evolution equation ($X$)? This is unconventional. Typically, the stochastic noise comes in through the velocity. By adding additional noise to the position statistics will not agree with the velocity statistics. By definition, a particle moves according to its velocity (see for example: Pope, 1994, Eq. 17).

4. Did the numerical solutions produce the so-called "rogue trajectories" (e.g., Yee and Wilson, 2007; Postma et al., 2012; Wilson, 2013), and if so, what was done to deal with them? And how might this affect results, particularly the energy spectra?

5. One problem with validating these subgrid models is that we often don't know what the correct answer is, and many times using no model is better than using a bad model. My opinion is that a more careful validation should be performed before making claims that the model is performing well. Based on the exercises presented, I do not feel such claims could be made. If the authors wish to test the model more closely, perhaps an *a priori* test, or even a toy problem may be a better means of testing the model.

Overall: With all of the above said, I'm not sure exactly what I would do to resolve all of these issues. Here is one suggestion. First, the authors should resolve the issues with the modeling and make it a true subgrid scale model. Then this could turn into a more applied study that is less about modeling methods and focuses more on the BLLAST experiment. I don't know all the measurements available from the BLLAST experiment (it is not well described in the paper), but perhaps you could force the LES with some larger-scale data and compare to some local measurements. Compare the two and discuss the successes/challenges.

**Minor Comments:**

1. Please number the equations.

2. Anytime an equation is added, it seems to mess up the line number count that follows.

**Appendix: Example MATLAB Code**

```matlab
clear

C0=4; %Kolmogorov's "universal" constant
C1=0.5+0.75*C0; %Pope's modified constant
eps=0.1; %turbulence dissipation rate
K=1; %turbulent kinetic energy

N=100000; %number of particles to be simulated
T=10; %length of simulation
dt=0.1; %time step

%initialize particle velocities using a normal distribution with unit
%variance.  NOTE: the initial values shouldn't matter provided T is
%large enough (particles should 'forget').
u=randn(1,N);

%march the velocity in time
for t=1:ceil(T/dt)
  u=u-C1*eps/K*u*dt+sqrt(C0*eps*dt)*randn(1,N);
end

%calculate the PDF from Lagrangian velocities
[P,U]=hist(u,15);
dx=U(2)-U(1);
P=P/sum(dx*P);%normalize so PDF integrates to unity

%"exact" Eulerian PDF
sigma2=K*C0/C1/2;
Gauss=1/sqrt(2*pi*sigma2)*exp(-U.^2/(2*sigma2));

%plot it
figure;hold on;
bar(U,P)
plot(U,Gauss,'-r')
xlabel('U')
ylabel('PDF')
legend('Lagrangian PDF','Eulerian PDF')
```

**References**

Cassiani, M., A. Stohl, and J. Brioude (2015). Lagrangian stochastic modelling of dispersion in the convective boundary layer with skewed turbulence conditions and a vertical density gradient: formulation and implementation in the FLEXPART model. *Bound.-Layer Meteorl. 154*, 367–390.

Pope, S. B. (1994). Lagrangian PDF methods for turbulent flows. *Ann. Rev. Fluid Mech. 26*, 23–63.

Postma, J. V., E. Yee, and J. D. Wilson (2012). First-order inconsistencies caused by rogue trajectories. *Boundary-Layer Meteorol. 144*, 431–439.

Shotorban, B. and F. Mashayek (2006). A stochastic model for particle motion in large-eddy simulation. *J. Turb. 7*, N18.

Vinkovic, I., C. Aguirre, and S. Simoëns (2006). Large-eddy simulation and Lagrangian stochastic modeling of passive scalar dispersion in a turbulent boundary layer. *J. Turb. 7*, N30.

Weil, J. C., P. P. Sullivan, and E. G. Patton (2004). The use of large-eddy simulations in Lagrangian particle dispersion models. *J. Atmos. Sci. 61*, 2877–2887.

Wilson, J. D. (2013). "Rogue velocities" in a Lagrangian stochastic model for idealized inhomogeneous turbulence. In J. Lin, D. Brunner, C. Gerbig, A. Stohl, A. Luhar, and P. Webley (Eds.), *Lagrangian Modeling of the Atmosphere*, pp. 53–57. Washington, DC: American Geophysical Union.

Yee, E. and J. D. Wilson (2007). Instability in Lagrangian stochastic trajectory models, and a method for its cure. *Boundary-Layer Meteorol. 122*, 243–261.

---

## Referee Comment (RC2) · Anonymous Referee #2 · 25 Mar 2016

In the manuscript "A new downscaling method for sub-grid turbulence modeling" present a proposal for a subgrid turbulence model based on a Lagrangian particle model. In this study the Lagrangian particle model is forced by resolved fields obtained from a large-eddy simulation (LES) with Meso-NH model. The focus is on analysis of small scale turbulence unresolved by Meso-NH model. Considering that the proposed subgrid model does not provide a feedback to the Meso-NH model this study can be considered as a-priori analysis of subgrid model.

General Remarks While Lagrangian particle models have been extensively used to simulate turbulent dispersion their application to turbulent flow modeling has been mainly in the area of combustion. The analysis presented in the manuscript can be considered as a priori analysis of subgrid turbulence without feedback to the resolved filed. The outline of the approach to turbulence modeling using new Lagrangian particle

model is lacking:

- the motivation for using Lagrangian particle model is not clearly stated,

- the focus is on assessment is on the subgrid turbulent kinetic energy (and spectra), however, for a model to be a viable turbulence model it needs to represent (subgrid) turbulent stresses and fluxes, but these were not evaluated,

- the analysis is limited to a relatively small part of the Meso-NH computational domain (2x2x2 coars grid cells, or 8x8x4 grid cells) and a short time period (15 minutes) and therefore the results may not be robust,

- the limitations of Lagrangian particle models have not been clearly stated and computational cost compared to more common turbulence closures has not been addresses,

- fundamental question that is not addressed in the manuscript is what scales of turbulence is Lagrangian particle model representing, this essential question needs to be addressed in the manuscript, and

- the conclusions that Lagrangian particle model follows Kolmogorov law is not supported by the results (i.e., the spectrum does not follow -5/3 scaling).

Considering that there may not be any previous examples of use of Lagrangian particle model as a closure for subgrid turbulence in LES (none are cited in the manuscript) the authors should have provided better background and motivation for their approach. Also, the Lagrangian particle model output is compared to higher-resolution LES, however, this is likely not a fair comparison considering that spatial and temporal resolution with Lagrangian particle model may be significantly higher than high-resolution LES. The authors do not address the issue of scales. Furthermore, the manuscript is not organized well: as indicated earlier, the motivation is not clearly stated, the model development is not concisely outlined, e.g, use of Meso-NH simulations to force the model is discussed before the model is presented, and the analysis does not follow logical order, e.g. turbulence spectra are analyzed before TKE is analyzed. Finally, the

manuscript in many instances does not follow proper English idiom as indicated below under "Specific Remarks." Taking all the above into account I do not recommend the manuscript for publication in the Journal Atmospheric Chemistry and Physics in the present form. The manuscript could be considered for publication after questions and suggestions for major revisions are addressed.

Specific Remarks

- Line 146, instead of "fine grid-point simulation" better would be "high-resolution simulation."

- Line 147 "The experience holds in three..." it should be "The process consists of three steps..."

- Line 151, same as above.

- Line 243, since the eddy dissipation rate is used in the model it should be stated how is it computed in Meso-NH.

- Line 254, instead of "close results" it should be "similar results."

- Line 258, instead of "experience" it should be "case," "study," or "experiment."

- Line 263, instead of "particle models" the terminology should be more precise: "Lagrangian particle models."

- Line 275, since the reader cannot know about authors' experience instead of "our experience" it should be "our study" or "our simulations."

- Line 323, number of particles used (75 per grid cell) is quite large, increasing grid resolution for the same number of grid cells per coarse grid cell would result in almost an order of magnitude higher resolution.

- Line 360, since dissipation of TKE (or more precisely energy transfer) is one of the critical roles of a subgrid model it would be important to describe how is energy dissipation rate modeled.

- Line 388, instead of a reference to Shannon (1949) original work by Nyquist should be referenced (This is actually Nyquist frequency, c.f. Certain factors affecting telegraph speed (1924) and Certain topics in Telegraph Transmission Theory (1928)).

- Line 405, "the aim of this work" should have been stated in the introduction.

- Line 427, the rationale for different treatment of horizontal and vertical velocities should be provided.

- Line 451, instead of ". . . we remind first how the TKE is computed..." better would be "...we first review how TKE is computed. . ."

- Figure 5, it is not clear what is the purpose and value of the comparison shown in this figure.

- Lines 509-514, the statements are based solely on a limited qualitative analysis (comparison of plots) and as such they are of little value.

- Line 523, the statement "more turbulent" is qualitative, that needs to be qualified. The question is what should be the level of turbulence at the scales that are resolved. Another question is what scales are particles representing?

---

## Author Comment (AC1) · 13 May 2016

First, we wish to thank the reviewer for his/her careful review. Below is our response to the comments (in blue) on a point-by-point basis. The text referring to the article is indicated in italic.

**Answers to major comments**

**1** The methods described in this paper are not a "sub-grid turbulence model". This is simply a RANS Lagrangian particle dispersion model. Essentially, the authors average out the grid-scale turbulent motions, and instead use the mean fields to drive the Lagrangian dispersion model (at least, this seems to be how it is described in the text). Thus, the mean field is the 'resolved' component and the Lagrangian evolution equation is used to add in the 'fluctuating' component, or in other words the authors are using a Reynolds decomposition :

$$u_i =  + u_i' \tag{1}$$

where $$ is an ensemble average, and $u_i'$ is the fluctuation from that average. If I am understanding the paper correctly, the authors then prescribe $$ using the Eulerian simulations, and $u_i'$ is (indirectly) determined using the Lagrangian stochastic evolution equation.
In order for this to be considered a "subgrid scale" model, the following decomposition would be used

$$u_i = \tilde{u}_i + u_i'' \tag{2}$$

where now $\tilde{u}_i$ is the filtered component (filtered at the grid scale), and $u_i''$ is the unresolved or subfilter component, which would be determined by solving the evolution equation in terms of the subfilter component :

$$du_i = a_0 dt - a_1(u_i - \tilde{u}_i)dt + b \ dW \tag{3}$$

where $dW$ is an increment in a Weiner process, and coefficients $a_0$, $a_1$, and $b$ are determined such that Eq. 3 satisfies the Navier-Stokes Equations. For examples of determining the coefficients see Weil et al. (2004), Shotorban and Mashayek (2006), or Vinkovic et al. (2006). These examples are applied to traditional LES applications, but the principles are the same as in the present manuscript. Thus, the turbulence calculated at the grid-scale is retained, and modeling is reduced to specifying only the subgrid scale turbulence. Why filter out valuable resolved turbulence by averaging ? The point of subgrid scale modeling is to retain as much information as possible, such that modeling is simplified in that we only have to model the smaller, more 'universal' scales.

The suggested downscaling method is derived from the Particle-In-Cell methods developed by Harlow [3] and used, for instance, in plasma modeling [6]. These methods combine Eulerian grid-point models and Lagrangian particles to model sub-grid fields. The downscaling method uses the same approach : large scales are resolved by an Eulerian model, and Lagrangian particles inside the grid cells are used to model sub-grid scales.
Within this framework, the article aims at presenting two different points : first a new stochastic downscaling method and then an application of the method to a BLLAST case. The downscaling method has been designed to model sub-grid

turbulence starting from a grid point simulation. The general idea is to force a sub-grid particle system with the resolved components of the grid point fields. In addition, to model the unresolved / sub-grid components, a stochastic Lagrangian model is used. For turbulence modeling, we suggest to use the following model, detailed in the article :

$$X_{k+1} = X_k + \mathbf{V}_k \; \delta t + \sigma^X \Delta B_{k+1}^X \quad (4)$$

$$V_{k+1} = V_k - \nabla_x \overline{p} \; \delta t \; - \; C_1 \frac{\varepsilon_k}{K_k} \left[ V_k - <V> \right] \delta t + \; \sqrt{C_0 . \varepsilon_k} \; \Delta B_{k+1}^V \quad (5)$$

$$W_{k+1} = W_k + \Delta_k W \; - \; C_1 \frac{\varepsilon_k}{K_k} \left[ W_k - <W> \right] \delta t + \frac{g}{\beta} \Gamma_k^\theta \; \delta t + \sqrt{C_0 . \varepsilon_k} \; \Delta B_{k+1}^W \quad (6)$$

where the control parameters (in blue) are given by the grid point fields.

Then the downscaling method has been applied to LES simulations performed for the BLLAST experiment. The ideal approach should apply the method to LES simulations on a coarse grid and then validate the sub-grid simulations by comparison with finer simulations of the same case performed with the same model. Unfortunately, this was not possible. Because of the high computational cost of the LES simulations performed with the Meso-NH model, simulations were available for only one grid size. These available simulations have been chosen as the reference simulations. Instead of simulations performed on a coarser grid, we have chosen to average the reference simulations on several grid points following a methodology used in [4]. The drawback of this solution is that the coarse fields include not only the components resolved on the grid, but also the average of the sub-grid components, being similar to RANS Lagrangian particle model. To conclude the answer to the first comment, we suggest to present the method and the application separately. The limits of the application are also now underlined.

Lines 142-146 : *The particle systems are forced with a large scale grid-point meteorological fields. The large scale used in our work is later described in details. The forced particles are used to model the sub-grid fields for the large scale model. To validate the sub-grid modeling a higher resolution model is used. In theory, the turbulent fields represented by the particles should be compared to the same fields simulated by a high resolution grid-point simulation. For computational reasons we do not have access to different high resolution simulations. Thus the large scale simulations have been built from the available simulations.*

Lines 155-161 : *Now that we have a reference simulation, a coarser simulation is built in order to force the particle system. To this end, we have chosen to average the grid-point fields on few cells. To be consistent, we have also applied a temporal average. The obtained coarse Meso-NH fields have thus lower spatial and temporal resolutions than the reference simulation while being consistent with it. However, due to the average, the coarse fields include not only the components resolved on the grid, but also the average of the sub-grid components. This limitation will be discussed section 8.*

Caption of figure 1 : *The downscaling experience. Coarse fields are used to force a sub-grid particle system. The sub-grid fields are compared to a reference Meso-NH simulation.*

**2** The authors state that the novelty of the proposed modeling methodology is that (Lines 77-78) "The method we suggest differs from these previous works : no assumption is made on the pdf shape." This is not true. By using a Langevin-based equation for the particle motions that is forced by a Weiner process, the authors are effectively specifying the shape of the pdf. Although Sects. 2.2 and 2.4 of Pope (1994) (for example) states that these methods do not assume a pdf, the form used by the author does assume a pdf. Particularly, it is assumed that the pdf is Gaussian with variance of $\sigma^2 = \frac{C_0 K}{2C_1}$ (where of course $K$ is the turbulent kinetic energy). The authors can verify this for themselves. Take the Lagrangian evolution equation, simulate some particles, and calculate the pdf. I have attached some sample MATLAB code to do this in an appendix of this review. For simplicity I have assumed that $\epsilon$ and $K$ are constant, and that there is no mean drift (i.e., $\nabla p = < U > = 0$). Means could be included, but they should only shift the pdf and not affect its shape. We should find that an ensemble of particles should have a velocity pdf of

$$P = (2\pi\sigma^2)^{-1/2} \exp(-\frac{u^2}{2\sigma^2})$$

where again $\sigma^2 = \frac{C_0 K}{2C_1}$ is the prescribed velocity variance. As a side note, this is not exactly true for the authors' equation for $W$ given the arbitrary way that they have included buoyancy (i.e., it is random and thus will generate erroneous additional variance). If it was correctly modeled, it should create a skewed velocity distribution and thus none of the analysis above or in the manuscript is valid. See Cassiani et al. (2015) and the references therein for examples.

We agree that a Langevin-based equation assumes the shape of the velocity pdf. The stochastic Lagrangian model that we used (equations 4 to 6) describes actually a Gaussian pdf when it is applied to a whole particle system with the same forcing and when $< \cdot >$ is the ensemble average over the particle system. The question becomes more complex when the model is locally applied, that is to say when the forcing varies in space and the average $< \cdot >$ is computed using only a subset of the particle system. In such case, the pdf is more difficult to describe.

The difference between the given MATLAB code and our code is that in our case, the forcing varies from one coarse cell to an other. Besides, the average $< \cdot >$ is a local average computed over the particles around the point of calculation, and the particles evolves freely in the whole domain. Therefore, at a given time in a given cell, particles with different characteristics are mixed, and pdf are only locally Gaussian. It leads that, the velocity pdf described by the particles in one fine cell is obtained by mixing Gaussian pdf but it is not necessarily Gaussian. To illustrate this, we have plotted on figure 1 the histograms of the first component of the velocity for one fine cell and for five consecutive time steps. One can see that the pdf may be far from the Gaussian distribution.

In the article, we will clearly detail this point. The misleading sentence "no assumption is made on the pdf shape" will be replaced by an explanation about the initial Gaussian assumption. A discussion about its validity in our case will also be added.

Lines 76-81 : *Nowadays, this kind of approximation is still widely used for subgrid modeling [7, 9, 8, 5, 2]. The method we suggest differs from these previous*

*works : the Gaussian assumption on the pdf shape is only locally made. Thus the particles give access to a discrete representation of the pdf which is not necessarily Gaussian at the scale of the grid cell. The pdf time evolution is thus given by the particle evolution.*

*Following line 782 : Concerning the SLM, another point has to be discussed. The Wiener processes used for the dispersion terms involve a locally Gaussian assumption of the pdf described by the particles. In our work, the Gaussian assumption is not valid at the grid cell scale. Indeed, at a given time in a given cell, particles with different characteristics are mixed. This is partially due to the free evolution of the particles in the domain. Thus the velocity pdf described by the particles in one fine cell is obtained by mixing Gaussian pdf but it is not necessarily Gaussian.*

[Figure]

Fig. 1: Histogram of the first component of the velocity (U) at five time steps for one cell.

**3** In the (unnumbered) equations following line 348 (note that line numbers are messed up here, there are more than 5 lines between labels 335 and 340), why is an additional noise term added to the position evolution equation $(X)$? This is unconventional. Typically, the stochastic noise comes in through the velocity. By adding additional noise to the position statistics will not agree with the velocity statistics. By definition, a particle moves according to its velocity (see for example : Pope, 1994, Eq. 17).

The added term in the particle location has been included to take into account the velocity integration errors considering the Euler scheme used for the velocity equation.

Indeed the stochastic Lagrangian model (SLM) is a McKean-Vlasov equation. $dV_t = b(V_t, \eta_t)dt + g(V_t)dW_t$, where $\eta_t$ is the $V_t$ probability law. As for Itô processes a Euler integration scheme is the most convenient. For Itô processes a Stratonovitch representation is possible, but for the McKean Vlasov, due to the mean term, a Euler scheme is most adapted.

The errors of Euler scheme for such processes are known (see [1]). These scheme errors are in root square of the time step $\sqrt{\delta t}$.

In the position evolution, the errors on the velocity evolution has to be taken into account. The location is the first integration of the velocity and a error term is added to the location : $X_{(k+1)\Delta t} = X_{k\delta t} + V_{k\delta t}\Delta t + \text{errors}$.

The error term is a stochastic processes taking into account the path dispersion. With errors in $\sqrt{\delta t}$-like, we choose a Wiener process to represent the dispersion process and $X_{k+1} = X_k + V_k \, \delta t + \sigma^X \Delta B_k^X$ where $\Delta B_k^X$ is a standard Wiener process.

The choice of the variance $\sigma^X$ is not anecdotal. We have already performed (unpublished work) a sensitivity analysis, and the variance $\sigma^X$ is of importance.

For the work presented in this article, $\sigma^X$ has been settled to 1.

Following line 346 : *One can notice the noise term at the end of the position evolution equation. The added term in the particle location has been included to take into account the velocity integration errors considering the Euler scheme used for the velocity equation (see [1]).*

**4** Did the numerical solutions produce the so-called "rogue trajectories" (e.g., Yee and Wilson, 2007 ; Postma et al., 2012 ; Wilson, 2013), and if so, what was done to deal with them ? And how might this affect results, particularly the energy spectra ?

The comment about rogue trajectories raises an interesting point. For dispersion models, one must ensure that modeled trajectories are coherent with the surrounding flow. In our case, particles are used to carry information about the modeled fluid. We are more interested in following the information than in following the particles themselves. Thus a specific particle management system has been designed to ensure that the particle density in each cell is high enough – a lack of particles would lead to a low quality modeling of the sub-grid fields. Because of this management system, particle trajectories are difficult to follow. Indeed, in average, particles are displaced every 3 or 4 time steps by the management system. Thus the trajectories are too short to be rogue. If we look at the longest trajectory without displacement observed during the simulation, we can see that it seems quite coherent with the flow (figure 2).

Lines 445-459 : *As the particles evolve freely in the domain, we also have to ensure a homogeneous repartition of the particles inside the domain. To do so, for each cell of the fine grid we keep the particle number between a minimal value and a maximal value which are given at the beginning of the simulation. By displacing particles, this method of particle management limits trajectory length and prevents rogue trajectories described by [12, 10, 11].*

[Figure]

Fig. 2: The longest particle trajectory (45 time steps of 5 s).

**5** One problem with validating these subgrid models is that we often don't know what the correct answer is, and many times using no model is better than using a bad model. My opinion is that a more careful validation should be performed before

making claims that the model is performing well. Based on the exercises presented, I do not feel such claims could be made. If the authors wish to test the model more closely, perhaps an a priori test, or even a toy problem may be a better means of testing the model.

As said in the comment, a major problem with validating the sub-grid model is that the correct answer is often unknown. In the present application, the reference is a high resolution grid-point simulation which is not directly used to force the particle system. One of the limitations for the validation has been that the resolution of the reference was not high enough. To tackle it, the ideal solution would be to use direct numerical simulations (DNS) of turbulent atmosphere to validate the sub-grid modeling. Such simulations are very costly and could not be done for the present case.

Concerning the use of toy models, we underline that a model sophisticated enough to model atmospheric turbulent flows is needed. That is why the Meso-NH model has been chosen. We agree that the downscaling method could be applied to an idealized case. However, at the beginning of this work, the resolution problem was not known and a real case –documented by turbulence observations and other studies– seemed a good application case to test the method.

The use of DNS or of a toy model would be interesting perspectives to complete this preliminary work. They should be added to the discussion part of the article. We will also precise that the article presents a first work and not a complete validation of the downscaling method. We suggest the following modifications :

Lines 761-764 : *This article presents a first work on a new way to model sub-grid processes using particle systems. One of the major improvement is the use of a simple turbulence model instead of complex model such as LES or DNS. However, to fully validate the method, one of the first steps should be to use a DNS or to apply the downscaling method to a toy model to know exactly the sub-grid fields. Unfortunately, such a validation could not have been done yet.*

**Références**

[1] Vlad Bally and Denis Talay. The law of the euler scheme for stochastic differential equations. *Probability theory and related fields*, 104(1) :43–60, 1996.

[2] Peter A Bogenschutz and Steven K Krueger. A simplified pdf parameterization of subgrid-scale clouds and turbulence for cloud-resolving models. *Journal of Advances in Modeling Earth Systems*, 5(2) :195–211, 2013.

[3] Francis H Harlow. The particle-in-cell method for numerical solution of problems in fluid dynamics. Technical report, Los Alamos Scientific Lab., N. Mex., 1962.

[4] Rachel Honnert, Valéry Masson, and Fleur Couvreux. A diagnostic for evaluating the representation of turbulence in atmospheric models at the kilometric scale. *Journal of the Atmospheric Sciences*, 68(12) :3112–3131, 2011.

[5] A Jam, F Hourdin, C Rio, and F Couvreux. Resolved versus parametrized boundary-layer plumes. part iii : Derivation of a statistical scheme for cumulus clouds. *Boundary-layer meteorology*, 147(3) :421–441, 2013.

[6] Giovanni Lapenta. Particle in cell methods.

[7] Vincent E Larson, Jean-Christophe Golaz, and William R Cotton. Small-scale and mesoscale variability in cloudy boundary layers : Joint probability density functions. *Journal of the atmospheric sciences*, 59(24) :3519–3539, 2002.

[8] Vincent E Larson, David P Schanen, Minghuai Wang, Mikhail Ovchinnikov, and Steven Ghan. Pdf parameterization of boundary layer clouds in models with horizontal grid spacings from 2 to 16 km. *Monthly Weather Review*, 140(1) :285–306, 2012.

[9] Emilie Perraud, Fleur Couvreux, Sylvie Malardel, Christine Lac, Valéry Masson, and Odile Thouron. Evaluation of statistical distributions for the parametrization of subgrid boundary-layer clouds. *Boundary-layer meteorology*, 140(2) :263–294, 2011.

[10] John V Postma, Eugene Yee, and John D Wilson. First-order inconsistencies caused by rogue trajectories. *Boundary-layer meteorology*, 144(3) :431–439, 2012.

[11] John D Wilson. rogue velocities in a lagrangian stochastic model for idealized inhomogeneous turbulence. *Lagrangian Modeling of the Atmosphere*, pages 53–58, 2013.

[12] Eugene Yee and John D Wilson. Instability in lagrangian stochastic trajectory models, and a method for its cure. *Boundary-layer meteorology*, 122(2) :243–261, 2007.

---

## Author Comment (AC2) · 13 May 2016

First, we wish to thank the reviewer for his/her careful review. Below is our response to the comments (in blue) on a point-by-point basis. The text referring to the article is indicated in italic.

**Answers to major comments**

**1** The motivation for using Lagrangian particle model is not clearly state.

To model wind fields at high temporal and spatial resolutions from a grid point simulation, there are two main methods. First, a method of adaptive grid refinement may be used. The method aims at refining the grid only where studied processes take place to be sure to have high resolution enough to resolve these processes. It is a costly method which requires the implementation of finite volume or finite element models. The alternative method is to use mesh-free particle models inside the grid cells. Starting from an Eulerian grid point model, the sub-grid fields are modeled using a Lagrangian approach. One of the main advantages of Lagrangian particle models is the ability to model nonlinear evolutions. This method is easier to implement than the adaptive grid refinement method, but it has its own drawbacks depending on the mesh-free model (interpolation issues for instance [4]). In this work, the mesh-free method has been chosen for a practical reason. The work on the Lagrangian model presented in the article has started from instrumental issue – signal filtering and turbulence estimation. In this framework, the Lagrangian particle model has given very good results [2, 12, 9, 10, 11]. For instance, using particle method, real time turbulence estimations are now available from wind observations. Thus using the same model for turbulence modeling was a natural approach. It has led us to a new downscaling method which opens new perspectives such as sub-grid TKE estimation in aeronautical areas or in data assimilation.We have modified the text in order to better state the motivation for using Lagrangian particle model.

Lines 62-81 *Instead of a larger decrease of the grid size, we suggest here another way to model sub-grid turbulence. In this paper, we present a stochastic downscaling approach. Our method is based on particle systems that are driven by a local turbulence model. Those particles are embedded in grid cells (illustration 1). [...] The pdf time evolution is thus given by the particle evolution. The suggested particle approach enables to model physical phenomena with nonlinear temporal evolutions. However, depending on the particle model, particle methods may have drawbacks such as interpolation issues for instance [4]. In the present work, the only delicate point is to ensure that the particle density is high enough in each grid cell.*

**2** The focus is on the assessment of the subgrid turbulent kinetic energy (and spectra), however, for a model to be a viable turbulence model it needs to represent (subgrid) turbulent stresses and fluxes, but these were not evaluated.

The focus is on the assessment of the subgrid turbulent kinetic energy because this variable is used in the Lagrangian particle model. Reynolds stresses $\overline{u_i' u_j'}$ may also be computed. It would be interesting to look at turbulent stresses and fluxes. However, turbulent fluxes cannot be computed for the particle fields since we have only wind simulations, and turbulent stresses are not usual studied variables to work

on convective boundary layer. Thus they have not been presented in the manuscript because it was a first work to test the downscaling method.

**3** The analysis is limited to a relatively small part of the Meso-NH computational domain (2x2x2 coarse grid cells, or 8x8x4 grid cells) and a short time period (15 minutes) and therefore the results may not be robust.

The small computational domain and the short time period are due to memory allocation problems of our programming software. From the presented study, we can not conclude about the result robustness. However, we have looked at the grid-point fields in areas of subsidence and of updraft. The fields were similar to the fields used in the present work, so the studied case may be considered as representative of the simulation. In addition, before using the BLLAST experiment simulations, simulations performed for the IHOP_2002 experiment have been used. These simulations have a larger grid size (100m for the horizontal resolution for instance [5]), this is why we have then worked on the BLLAST simulations. The very first results obtained with the IHOP_2002 simulations were very similar to the presented results indicating some universality of the results for convective boundary layer. Thus, even if the robustness has still to be proved, we are rather confident about it.

Lines 765-774 *This experience has been realized on a small domain, with a reduced number of particles in each cell. These two constraints were related to the long computational time. Extending the domain and the duration of the simulation should be one of the next steps. It would improve the PSD quality, and limit the influence of the edges. Then, a supplementary work on the spatial resolution of the particle simulations might be done and the result robustness may be tested. To give a very first answer to the robustness issue, we remind that we have compared the studied fields to fields on different areas on the same vertical level. The comparison has shown that Meso-NH fields are similar in the different areas. Thus, the downscaling method should provide similar results when being applied in these areas. Extending the duration of the simulation would also enable us to compare the particle wind and the particle TKE to the in-situ observations.*

**4** The limitations of Lagrangian particle models have not been clearly stated and computational cost compared to more common turbulence closures has not been addresses.

We underline that the aim of this work is not to provide cheap turbulence closures. The idea is instead to work on a new way to model sub-grid turbulence for research purposes. On one hand, particle methods are costly, and require a particle management method to ensure that there are enough particles in each grid cell to compute sub-grid fields. On the other hand, particle methods are adapted to model nonlinear physical processes, such as atmospheric turbulence. Besides, horizontal and vertical velocities are treated differently because the Lagrangian model does not assume homogeneous or isotropic turbulence. Thus the new downscaling method should be seen as an alternative solution to common turbulence closures which often assume isotropic and homogeneous turbulence. In the long term, it could also be seen as a tool to improve or to evaluate the quality of common turbulence closures used in operational models.

Lines 765-774 *Among the future works, there is first the application of the down-scaling method to a larger domain, and the comparison of the sub-grid fields to observations. Then, there is the study of the TKE parametrization used in Meso-NH by comparison with the TKE modeled by the particles. Despite the computational time, in a long-term perspective we may also think to experiences where the sub-grid parametrization used in Meso-NH will be replaced by sub-grid particle modeling. Indeed, for research purposes, the downscaling method may be an alternative solution to common turbulence closures which often assume isotropic and homogeneous turbulence.*

**5** Fundamental question that is not addressed in the manuscript is what scales of turbulence is Lagrangian particle model representing, this essential question needs to be addressed in the manuscript.

The question of the scales represented by the Lagrangian particle model is actually a fundamental question. So far we have few answers to this question. To know the spatial scale associated to each particle, a Voronoï diagram could be plotted using particles as seeds. However it is a costly solution to know the particle scale, and we are not sure that the physical interpretation of this scale is relevant. A cheaper solution is to divide the grid cell surface by the mean number of particles. There are 75 particles in each 40mx40m cell, that is to say that each particle represents a 21m$^2$ surface. Thus a particle has approximatively a length of 4.6m. An other solution to know the scales represented by the particle system is to use the power spectrum densities and the mean velocities to evaluate Lagrangian lengths. Looking at the spectrum of the first component of the wind, we can see that the spectrum is flat for frequencies higher than $5.10^{-2}\text{s}^{-1}$ (figure 1). In average over the domain, the first component of the particle wind is about 1.3m/s. Thus a Lagrangian length associated to the particles for the first wind component is about 26m. The second and the third mean wind components are 0.3m/s and -0.22m/s. Thus, using the same cut-off frequency, we obtain Lagrangian lengths about 6m and 4m for the second and the third wind component respectively.
So far these lengths are the only estimations of the scales represented by the Lagrangian particle model. However a specific work has still to be done to figure out the scales represented by the Lagrangian particle model. To underline this issue, paragraphs will be added in the section which presents the power spectrum densities and in the discussion section.

Before line 622 *The spatial resolution of the particle simulations is tricky to estimate. A first estimation may be given by the Lagrangian lengths associated to the wind components. The lengths can be evaluated using the power spectrum densities and the mean velocities. Looking at the spectrum of the first component of the wind, we can see that the spectrum is flat for frequencies higher than $5.10^{-2}s^{-1}$ (figure 8). In average over the domain, the first component of the particle wind is about 1.3m/s. Thus a Lagrangian length associated to the particles for the first wind component is about 26m. Using the same cut-off frequency, we obtain Lagrangian lengths about 6m and 4m for the second and the third wind component respectively. The differences*

[Figure]

Fig. 1: Power spectrum densities of the components of the fine Meso-NH wind in black and of the components of the particle wind in red, calculated on a 4x4 grid cells domain, for the second level of the fine grid. In green we have the -5/3 slope according to the K41 laws.

*between the lengths computed for the three components clearly show the difficulty to use this method to evaluate the spatial resolution of the particle simulations.*

Following line 774 *Related to the question of the spatial resolution of particle simulations, there is also the fundamental question of the scale of the turbulence represented in the particle fields. So far, only a first estimation of the scale has been given, and a specific work has still to be done to figure out the scales represented by the particle model.*

**6**   The conclusions that Lagrangian particle model follows Kolmogorov law is not supported by the results (i.e., the spectrum does not follow -5/3 scaling).

As explained in the article, the Stochastic Lagrangian Model comes from the work of S.B. Pope [8]. The original model presented by Pope has been designed to follow the Kolmogorov law. Then it has been adapted for atmospheric turbulence estimation by Baehr [1, 2]. In the turbulence estimation framework, results have shown that the model follows the Kolmogorov law. The present downscaling method uses the model from Baehr work with a small simplification due to the vertical size of the domain – we assume that vertical gradients of horizontal wind are null. The main difference with the works of Pope and Baehr is not about the model formulation. It is due to the different ways to compute the control parameters. Our work is a first attempt to use control parameters given by an Eulerian grid point simulation. In this framework, the model behaviour has not been completely assessed

yet. As presented in the article, the spectra do not perfectly follow the -5/3 scaling. However, they have regular slopes. One can see that it is not the case for the Meso-NH spectra. Longer simulations are needed to continue working on the model behaviour. Spectrum calculated from longer time series will be more representative of the model behaviour, and the spectrum shape will be easier to interpret. It has not been done yet because of computational costs. Another interesting point is to work on one of the control parameters, the eddy dissipation rate (EDR). As it is a preliminary work, the EDR variable of Meso-NH has been used. This variable is computed using a closure scheme which can introduce errors on the EDR modeling. An alternative solution could be to compute the EDR directly from the grid point wind fields. Improving the EDR modeling should directly improve the sub-grid turbulence modeling and thus the model behaviour.

Lines 593-604 *Figure 9 presents the PSDs of the three components of the fine Meso-NH wind and of the particle wind. It appears that neither the particle wind or the fine Meso-NH wind follows perfectly the energy cascade given by the Kolmogorov's theory and represented by the -5/3 slope. But we may notice that the particle wind spectra present a regular slope. The regularity of the slope is a good point to assess the particle wind and its fast fluctuations. It shows that the energy cascades are the same whatever the considered scales. However the slope is slightly more gentle than the energy cascade. There may be too much energy associated to the turbulence modeled by the particles at high frequencies. As presented in section 3.4, the SLM has been designed to follow Kolmogorov laws, but the spectra on figure 9 have been obtained by applying the model in a new framework. Our work is a first attempt to use control parameters given by an Eulerian grid point simulation. In this framework, the model behaviour has not been completely assessed yet. Longer simulations are needed to continue working on it.*

**Manuscript organization**

Considering that there may not be any previous examples of use of Lagrangian particle model as a closure for subgrid turbulence in LES (none are cited in the manuscript) the authors should have provided better background and motivation for their approach.

The motivation for our approach has been detailed in the first point of the "Answers to major comments" section. Modifications in the article have also been suggested in this section.

Also, the Lagrangian particle model output is compared to higher-resolution LES, however, this is likely not a fair comparison considering that spatial and temporal resolution with Lagrangian particle model may be significantly higher than high-resolution LES.

We agree that spatial and temporal resolution with Lagrangian particle model may be significantly higher than high-resolution LES. However we have done the comparison because it was the only way to study the sub-grid fields modeled using a the particle system. In the present application, a high resolution LES has been used as a reference. This LES has been chosen because the simulated case was documented by turbulence observations in an other study [7]. In order to force the

particle system we have built a lower resolution simulation using the reference LES. One of the limitations for the validation has been that the difference of resolution was not large enough. Thus the resolution of the reference was not high enough to be compared to the particle simulation. To tackle it, the ideal solution would be to use direct numerical simulations (DNS) of turbulent atmosphere to validate the sub-grid modeling. Such simulations are very costly and could not be done for the present case.

Lines 761-764 : *This article presents a preliminary work on a new way to model sub-grid processes using particle systems. One of the major improvement is the use of a simple turbulence model instead of complex model such as LES or DNS. However, to fully validate the method, one of the first steps should be to use a DNS or to apply the downscaling method to a toy model to know exactly the sub-grid fields. Unfortunately, such a validation could not have been done yet.*

The authors do not address the issue of scales.
Concerning the interesting issue of scales represented by the Lagrangian particle model, two paragraphs will be added in the article (see point 5).

Furthermore, the manuscript is not organized well: as indicated earlier, the motivation is not clearly stated, the model development is not concisely outlined, e.g, use of Meso-NH simulations to force the model is discussed before the model is presented, and the analysis does not follow logical order, e.g. turbulence spectra are analyzed before TKE is analyzed.
Motivations will be clearly indicated and models will be presented before discussing the use of Meso-NH simulations to force the particle system. Concerning the TKE analysis, we have chosen to present it after the analysis of the wind and the power spectrum densities because the subgrid TKE is computed from the particle TKE and not directly from the wind (see section 5.2 of the article). Thus we have chosen to analyse all results about the sub-grid wind (including spectra) before analysing results about the sub-grid TKE. An explanation of this choice will be added at the beginning of the results section.

Lines 482-488 *In the previous sections, the downscaling algorithm has been described in details. The obtained results are now presented. To assess the behavior of the particle system, we compare the 3D wind given by the particle and by Meso-NH. First results on the coarse grid are shown. Then we present the comparison between the particle fields and the fine Meso-NH fields. Wind power spectrum densities are then presented. Finally, results for subgrid TKE are presented. This results are presented separately because the subgrid TKE is computed from the particle TKE and not directly from the wind.*

**Specific remarks**

- Corrections about the vocabulary will be taken into account. The authors thank the referee for the suggestions.

- Line 243, since the eddy dissipation rate is used in the model it should be

stated how is it computed in Meso-NH.

The eddy dissipation rate is computed from the subgrid turbulent kinetic energy which is a prognostic variable of Meso-NH. More precisely, a closure scheme based on the mixing length is used [6].

Lines 201-210 *For our simulations, the 3D turbulence scheme is a one and a half order closure scheme (Cuxart et al., 2000). Thus the sub-grid TKE is a prognostic variable whereas the mixing length is a diagnostic variable. The mixing length and the dissipative length are computed separately according to Redelsperger et al. (2001). The mixing length is given by the mesh size depending on the model dimensionality. This length is limited to the ground distance and also by the Sommeria and Deardorff (1977) mixing length, which is pertinent in the stable cases. The eddy dissipation rate is computed from the subgrid turbulent kinetic energy using a closure scheme based on the mixing length.*

- Line 323, number of particles used (75 per grid cell) is quite large, increasing grid resolution for the same number of grid cells per coarse grid cell would result in almost an order of magnitude higher resolution.

The number of particles which has been used may seem large, but it may be compared to the number of particles (800 per grid cell) used by Bernardin et al for the same kind of application [3]. As there is a link between the number of grid cells and the resolution of the grid point model, there is a link between the number of particles and the Lagrangian particle model resolution. However we think that the number of particles may not be directly compared to the number of grid cells. As explained previously, the work on the Lagrangian particle model resolution has not been done yet.

Lines 321-323 *The results of the sub-grid modeling are presented in section 6. They are obtained with 75 particles in each fine grid cell. Thus the whole system contains 19200 particles. This number may be compared to the 800 particles per grid cell used by [3] for the same kind of application.*

- Line 388, instead of a reference to Shannon (1949) original work by Nyquist should be referenced (This is actually Nyquist frequency, c.f. Certain factors affecting telegraph speed (1924) and Certain topics in Telegraph Transmission Theory (1928)).

The reference to Shannon work will be replaced by a reference to Nyquist paper.

- Line 427, the rationale for different treatment of horizontal and vertical velocities should be provided.

The rationale for different treatment of horizontal and vertical velocities is that the velocities are not driven by the same physical processes. Large scale

horizontal velocities are driven by pressure gradients, whereas large scale vertical velocities are driven by buoyancy. To improve our study, temperature gradients computed from Meso-NH simulations could be taken into account.

Lines 427-432 *The horizontal velocities are forced with the pressure gradient and the dissipation rate. For the vertical velocities, the forcing is slightly different : it uses the vertical velocity coarse fields instead of the pressure fields. This choice has been done because horizontal velocities are driven by pressure gradients, whereas vertical velocities are driven by temperature gradients. To improve the downscaling method, temperature gradients computed from Meso-NH simulations could be taken into account. We may notice that in this work, the EDR used to force the particles is considered isotropic.*

- Figure 5, it is not clear what is the purpose and value of the comparison shown in this figure.

Figures 5 compares the wind modeled by the coarse grid point simulation to the particle wind averaged coarse cell by coarse cell. The aim is to assess the particle behaviour at the forcing scale. This step is necessary because, for the horizontal velocity, the average particle behaviour is given by the pressure gradient and not by an average velocity.

Lines 497-504 *To assess the downscaling method, the first thing to look at is the agreement between the coarse wind and the average of the sub-grid wind modeled by the particle system. The aim is to assess the particle behaviour at the forcing scale. This verification is important, especially for the horizontal velocity which is not directly forced by the coarse horizontal velocity fields. We remind that to compare the particle wind to the coarse wind fields, the particle values are averaged coarse cell by coarse cell.*

- Lines 509-514, the statements are based solely on a limited qualitative analysis (comparison of plots) and as such they are of little value.

The qualitative analysis may be completed with a more quantitative analysis. For instance root mean square errors (RMSE) can be added. For the first component, the RMSE of the particle wind averaged at the large scale is 0.045 m/s. For the second and the third components, the RMSE are respectively 0.062 m/s and 0.135 m/s.

Lines 507-514 *The averaged particle wind is consistent with the coarse wind, especially for the horizontal wind. The root mean square errors associated to the first and the second particle wind components are respectively $0.045 m.s^{-1}$ and $0.062 m.s^{-1}$. For the vertical wind, there is more discrepancy between the particle wind and the Meso-NH wind, but they present the same variations. The associated error is $0.135 m.s^{-1}$. Similar results have been obtained for the other coarse cells (not shown). Thus, as expected, the 3D wind modeled by the particles is in good agreement with the wind modeled by the coarse Meso-NH model.*

- Line 523, the statement "more turbulent" is qualitative, that needs to be qualified. The question is what should be the level of turbulence at the scales that are resolved. Another question is what scales are particles representing.

The statement "more turbulent" is implicitly qualified by the shape of the power spectrum densities represented figure 1 –figure 9 in the manuscript. Looking at the spectra, one can see that the energy associated to high frequencies is higher in the particle wind than in the reference Meso-NH wind. An explanation of the statement will be added to the manuscript. The complementary issue of the scales represented by the particles has not been addressed in the manuscript. As explained previously, it is still an open question. This preliminary work has raised questions about the effective resolution of the particle Lagrangian model. A more advanced work is necessary to answer it. We will add paragraphs about this issue in the article (see point 5).

Lines 523-527 *First, we notice the more turbulent profile of the 3D wind represented by the particles than the fine Meso-NH wind profile. Indeed, the Meso-NH wind appears smoother while the particle wind presents more temporal fluctuations. The interpretation of the power spectrum densities presented section 6.1.3 will confirm that the energy associated to high frequencies is higher in the particle wind than in the reference Meso-Nh wind.*

**References**

[1] C. Baehr. Stochastic modeling and filtering of discrete measurements for a turbulent field. application to measurements of atmospheric wind. *International Journal of Modern Physics B*, 23(28-29):5424–5433, 2009.

[2] C. Baehr. Nonlinear filtering for observations on a random vector field along a random path. application to atmospheric turbulent velocities. *ESAIM: Mathematical Modelling and Numerical Analysis*, 44(05):921–945, 2010.

[3] Frédéric Bernardin, Mireille Bossy, Claire Chauvin, Philippe Drobinski, Antoine Rousseau, and Tamara Salameh. Stochastic downscaling method: application to wind refinement. *Stochastic Environmental Research and Risk Assessment*, 23(6):851–859, 2009.

[4] Jeremiah U Brackbill, Douglas B Kothe, and Hans M Ruppel. Flip: A low-dissipation, particle-in-cell method for fluid flow. *Computer Physics Communications*, 48(1):25–38, 1988.

[5] F Couvreux, F Guichard, JL Redelsperger, C Kiemle, V Masson, JP Lafore, and Cyrille Flamant. Water-vapour variability within a convective boundary-layer assessed by large-eddy simulations and ihop_ 2002 observations. *Quarterly Journal of the Royal Meteorological Society*, 131(611):2665–2693, 2005.

[6] Joan Cuxart, Philippe Bougeault, and J-L Redelsperger. A turbulence scheme allowing for mesoscale and large-eddy simulations. *Quarterly Journal of the Royal Meteorological Society*, 126(562):1–30, 2000.

[7] C Darbieu, F Lohou, M Lothon, J Vilà-Guerau de Arellano, F Couvreux, P Durand, D Pino, E G Patton, E Nilsson, E Blay-Carreras, et al. Turbulence vertical structure of the boundary layer during the afternoon transition. *Atmospheric Chemistry and Physics*, 15(23):10071–10086, 2015.

[8] Stephen B Pope. *Turbulent flows.* Cambridge university press, 2000.

[9] L. Rottner and C. Baehr. 3D Wind Reconstruction and Turbulence Estimation in the Boundary Layer from Doppler Lidar Measurements using Particle Method. *AGU Fall Meeting Abstracts*, December 2014.

[10] L. Rottner, I. Suomi, T. Rieutord, C. Baehr, and S-E. Gryning. Real time turbulence and wind gust estimation from wind lidar observations using the turbulence reconstruction method. *18th International Symposium for the Advancement of Boundary Layer Remote Sensing*, 2016.

[11] I. Suomi, L. Rottner, E. O'Connor, S-E. Gryning, and A. Sathe. Wind gust measurements using pulsed Doppler wind-lidar: comparison of direct and indirect techniques. *22nd Symposium on Boundary Layers and Turbulence, American Meteorological Society*, 2016.

[12] Florian Suzat, Christophe Baehr, and Alain Dabas. A fast atmospheric turbulent parameters estimation using particle filtering. application to lidar observations. In *Journal of Physics: Conference Series*, volume 318, page 072019. IOP Publishing, 2011.

---

## Referee Report (RR1)

**Review of acm-2015-1015**

**Overall Comments:**

This review concerns the manuscript acm-2015-1015, which I reviewed previously. I have read through the authors' replies to both previous reviews. I still have some substantial concerns related to the overall methodology of the paper and some of the technical details. I tend to agree with the other reviewer's comments that the organization and presentation of the paper could be significantly improved. After reading through the paper quite a few times, I believe I may have a handle on the overall methods of the study. Based on my own (probably biased) opinion, I feel that the authors have made things much more complicated and confusing than they need to be. I would summarize the study as follows: A mesoscale CFD model (Meso-NH) is used to simulate flow in the atmospheric boundary layer at the kilometer scale. Since it is not computationally feasible to resolve the smallest eddies, the velocity provided by Meso-NH is essentially a 'filtered' velocity that is missing energy due to subgrid effects. The overall goal of this paper, is to develop a methodology that can give a statistical representation of the *total* velocity (resolved plus subgrid) on a grid with arbitrary resolution. For this task, they use a Lagrangian stochastic dispersion model to give the total Lagrangian velocity, which is aggregated over an Eulerian grid cell using a local averaging operator. The Lagrangian model is based on that of Pope, but it is essentially the same as that of Thomson to within a constant and minus the 'drift correction' terms. The unusual thing here is that it would appear that although there is some resolved turbulence in the Meso-NH solution, they apply a RANS Lagrangian model, which seems that it is inconsistent (more on this below). The authors also throw in some arbitrary buoyancy and truncation error correction terms, which in my opinion are inconsistent as they are ultimately just arbitrary Gaussian noise.

To test the model, the authors use Meso-NH to simulate a 15-hour portion of the BLAAST experiment on a grid of $256^3$ points. But since the authors argue that it is too computationally expensive to consider all this data, and instead they only consider a 15-minute period on an $8 \times 8 \times 4$ sub-set of the grid. The Lagrangian model is driven by filtering the $256^3$ "fine" grid simulation, to give a "coarse" grid simulation that is half the resolution. The Lagrangian model (driven by the "coarse" data) is validated by comparing back to the "fine" grid data. It is my impression that the authors feel that the velocity statistics from the Lagrangian downscaling model should match the Eulerian statistics on the "fine" grid. They of course find that the downscaling model adds in back in too much energy, but overall they conclude that "The particle wind seems in good agreement with the high resolution wind".

Overall, the problem seems to me to be that 1) the model is inconsistently formulated,

and 2) the "validation" methodology is ill-posed. Unless I am misunderstanding something, the solution to all of this is relatively straightforward, as much of this has been done in the past in different ways, the pieces just need to be put together. As detailed below, this involves 1) using a form of the Lagrangian model that is theoretically consistent, and 2) validation using a consistently-posed study. The comments below are not meant to be overly onerous or critical, but rather helpful. For transparency I would also note that stochastic downscaling of mesoscale data is not my research area, and thus I am not trying to "protect" my own work in any way (i.e., whether or not this work is published has no effect on my own work).

**Previous Review Major Comments:**

1. Sub-grid turbulence model: My previous comment was essentially that it appears that the model is inconsistent with behavior that is expected in a subgrid turbulence model. My claim was that the particles should be driven by the resolved Eulerian velocity field, and the Lagrangian model should be predicting the unresolved velocity.

Based on my understanding, the behavior of the model presented by the authors appears to be inconsistent with that of a subgrid-scale model. Perhaps this is because my understanding of the model formulation is still incomplete, in which case the authors may be able to explain its consistency through the following simple thought experiment. Simply consider how the model should respond as grid resolution is varied. As the grid scale approaches the Kolmogorov scale, the subgrid effects should tend toward zero (this is the most fundamental quality of a subgrid model). And in the context of the particle model, the particle velocity should match the Eulerian velocity: $V \to v$ and $W \to w$. Forgive me, but looking at the equation following Line 419 I don't see how this will happen. In the opinion of this reviewer, the authors either need to demonstrate that the model is at least consistent in this regard, or use a model formulation that is consistent.

I am still not entirely sure why the authors have set up their Lagrangian equations the way they have, rather than using a standard approach that is known to be consistent (it is no more difficult or costly). The authors are of course free to use any approach they like as long as it's consistent, I am just curious.

So why not calculate the total velocity $V$ as the sum of the resolved Eulerian grid velocity (known, call it $v$) and model the unresolved component using the SLM (unknown, call it $\tilde{v}$); i.e., $V = v + \tilde{v}$? Note that in the current paper, $v = -\nabla_x \overline{p}$ which is wrapped into the total equation for $V$, whereas here $v$ is interpolated from the Eulerian simulation to the particle location and thus is 'known'. Then the evolution equation for $\tilde{v}$ is left to model, which is

$$ d\tilde{v} = -C_1 \frac{\varepsilon}{e} \tilde{v} dt + \sqrt{C_0 \varepsilon} dB, \tag{1} $$

where $\varepsilon$ and $e$ are turbulent dissipation rate and subgrid TKE, respectively, which are interpolated from the Eulerian grid to the particle position. Now, our model will at least be consistent as the grid is refined: $e$ tends toward zero as the grid scale tends toward zero (as a result this dissipation term becomes very large and damps out all fluctuations), and thus $\tilde{v} \to 0$ and $V \to v$. This formulation is also consistent if you go the other way and tend toward RANS. As the grid scale becomes very large, $e \to K$, $v \to \overline{v}$, $\tilde{v} \to v'$, and thus we converge to a standard RANS downscaling model. This seems important for Meso-NH, which can be 'switched' between RANS and LES modes. Only in the case of RANS where no turbulence is resolved is the resolved velocity equal to $-\nabla_x \overline{p}$ and the unresolved TKE equal to $K$.

2. Gaussian assumption: Unfortunately, I am still in disagreement with the authors' position, as well as their newly added statements, e.g., "It leads that in this study, in a given grid cell, particles are samples of different Gaussian pdf. " It doesn't matter if the turbulence statistics vary in space, the particle velocities will be locally Gaussian with variance, dissipation, etc. equal to the Eulerian value specified at that point. This is the idea behind the Pope/Thomson Lagrangian stochastic dispersion models. We specify a PDF at every point, and we seek to generate an ensemble of Lagrangian particles that has that PDF and has a local dissipation rate of $\varepsilon$. This is how the models are derived, and how they work out in practice. Any deviation from this is caused by numerical error (or in the authors' case it could be due to the arbitrary stochastic terms that were added). The advantage of using an SLM rather than just sampling a PDF is that the particle velocities are correlated in time (this is a result of the fact that they must have dissipation rate of $\varepsilon$, or more directly that they follow Kolmogorov's second similarity hypothesis). Also as a side note regarding why the histograms presented by the authors (Fig. 1 of reply) do not appear Gaussian: the sample size is too small to make such an assessment. With so few samples (i.e., 75), the histograms are unlikely to look Gaussian unless we get lucky. The authors can test this using the MATLAB code I provided in the previous review, and setting N=75. In this case the PDF will usually look quite non-Gaussian (depending on 'luck'), and as N is increased the PDF converges toward Gaussian.

With all that said, I am not particularly concerned with this issue as it pertains to the manuscript. The original motivation for this comment was related to the novelty of the paper. Let's say for the sake of argument that the authors are correct that for some reason the particle velocities are highly non-Gaussian when aggregated over a grid cell. Even if that is so, what is the novelty of the downscaling methodology when considering the work of, e.g., Bernardin et al. (2009)? Looking at their Eq. 19b, the only difference I see between the authors' velocity equation is the arbitrary "buoyancy" term that was added. Based on my own viewpoint, I would say that the novelty lies in the fact that the authors have used the Meso-NH model, which based on my limited understanding, is somewhat of a hybrid between a RANS and LES code. In that case, it seems that the downscaling method should be consistent in that regard, and thus it might make more sense to use the

approach described in the previous comment.

3. Stochastic numerical error 'correction' term in particle position equation: Can the authors provide any references that 'correct' for the numerical errors in this way? I can't see how adding additional dispersion somehow corrects for the numerical dispersion – in this case I don't see how two wrongs make a right, as both the error and correction appear to be additive and dispersive. Typically, numerical truncation errors are reduced by reducing the timestep, or there are refinement methods that use a systematic approach to (usually iteratively) improve the solution. Maybe the authors are using some method that I am unfamiliar with, in which case I am curious to read more (perhaps from a reference).

Also, I would note that the 'correction' is being added to the position evolution equation, yet the explanation given by the authors seems to pertain to errors related to the velocity evolution equation. How are the two related?

4. Rogue trajectories: Overall, the authors' response was sufficient. I would recommend one thing the authors might try. What happens to the power spectra when you decrease the particle integration timestep $\delta t$ by, say, an order of magnitude? Does it get rid of some of that extra energy at small-scales? If so, there may be some issues with numerical stability although it might not be manifesting as 'rogue' trajectories in the traditional sense.

5. Validation: Even on the "fine" Meso-NH grid, the velocity field is presumably still missing unresolved (subgrid) energy, which I'm guessing is not negligible (looking at the spectra in Fig. 9, it appears to be substantial). This is why the Mesh-NH velocity is much "smoother" than the particle velocity. So the When you filter (i.e., average) to get the "coarse" grid, some additional energy is removed, let's call that $\Delta e$. So the total TKE for the coarse grid is $K_{res} + \Delta e + e$. Here is something to try: why not use $\Delta e$ instead of $e$ in Eq. 1 (above), and compare to the resolved TKE from the "fine" solution? In this case, the goal of the downscaling model is to recover $\Delta e$ rather than $\Delta e + e$, which means you can directly compare to $K_{res}$ for the fine grid. My opinion is that simply saying that for future work "higher resolution simulations should be performed" is not acceptable.

Regarding a "toy problem": I could think of some tests that could be useful here. What about generating some random 'turbulence' (could be white noise or correlated) on the "fine" grid, then filtering it to get a "coarse" grid? You could calculate the TKE of the fine grid turbulence (this is '$K = K_{res} + e$'), then calculate the TKE of the coarse grid (this is $K_{res}$). Now drive the particle model with $e$ and downscale to the fine grid, where you should find that the TKE of the total particle velocity is $K$. This of course is non-physical and probably wouldn't go into the paper, but could be a good verification check to demonstrate consistency.

**Minor Comments:**

1. In my own experience, the term 'downscaling' is usually used to describe a one-way model from large to small scales, whereas 'sub-grid modeling' is typically reserved for two-way coupling where the large-scale model needs to parameterize the small-scales. I would consider this work to address downscaling. I would leave it up to the authors discretion, but they may consider revising the title and certain other instances to make this point clear.

2. Line 25: model → models

3. Lines 41-44: Consider re-wording this sentence. How can AROME airport resolve processes? The authors probably mean that processes at the scale of AROME airport are not resolved.

4. Lines 80-86: The first statement seems to say that an assumption is made that the local PDF is Gaussian. Then it says that locally the PDF samples multiple Gaussian PDFs, and therefore it is not necessarily Gaussian. These seem to contract each other.

5. Sect. 3.2: What is meant by the term 'coupling experience'? In English, this phrase sounds a bit unusual. Is there some particular reason to use the word 'experience', rather than just saying something like 'model coupling', or 'coupling between resolved and unresolved scales'?

6. Sect. 3.3: Consider moving this section until after the model has been introduced (i.e., beginning of Sect. 6). Currently, it feels out of place since this is really just details related to model testing/validation and is not central to the model itself.

7. 400-403: Can the authors explain why they feel that the ideas presented by Kolmogorov are considered "laws"? Typically these ideas are referred to as 'Kolmogorov theory' or 'Kolmogorov's hypothesis', as they are largely based on similarity/scaling arguments.

8. 400-403: How exactly is the model consistent with Kolmogorov theory? Would the authors consider it to be consistent with all of the similarity hypotheses presented in K41, or is it that it is consistent with Kolmogorov's second similarity hypothesis in that the variance of the Lagrangian velocity increments ($\langle du^2 \rangle$) is proportional to the turbulence dissipation rate, i.e., $C_0 \varepsilon dt$?

9. Lines 419.5 (equation): I am not entirely clear on how $K$ is specified. Normally, this would come from the large-scale simulation and be interpolated to the particle position. On Lines 308-310, the authors mention that the TKE is extracted from Meso-NH, which would lead me to believe that is where $K$ comes from. However, Sect. 5.2 would suggest otherwise, that somehow the TKE is calculated afterword, although it is required in the velocity equation itself. Please explain.

10. Lines 423-426: The statement regarding the buoyancy term is vague. There are an infinite number of ways in which this term could be modeled using a random variable. Was this simply an empirical 'knob' that was turned?

11. Lines 504-505: How is the particle velocity initialized? This seems important considering that the particle simulation times ($\Delta t$) are much shorter than the integral time scale, and therefore they are likely to 'remember' the initial condition.

12. Sect 3.6: This title seems inappropriate. When are structure functions ever calculated in the paper? It seems like a more appropriate title might be 'Ensemble averaging' or something of that nature.

13. Lines 635-651 and Figure 7: I don't see the value of this comparison. Firstly, the simulations and the data are not over the same time period. Secondly, am I supposed to look at Fig. 6 and Fig. 7 and say "Yes, the Meso-NH velocity is smoother than the sonic data"? It would be incredibly surprising if that were not the case considering that the simulations don't resolve below the grid scale.

14. Lines 671-674: Should we not expect better agreement if the method is consistent?

15. Lines 855-858 and Fig. 12 caption: It is not clear to me how exactly the 'new' grid was obtained. If the grid resolution is changed, shouldn't the 'Meso-NH' TKE (black line) change as well?

16. Sect. 6.1.3 (velocity spectra): I have quite a few concerns with this section:

- For the SLM used here, we know that 1. An ensemble of particles at any location should have TKE $K$, and 2. The variance of particle velocity increments at any location should be $\langle du^2 \rangle = C_0 \epsilon dt$, which is consistent with Kolmogorov theory. Given this, should we expect the particle velocity spectra to follow $k^{-5/3}$ scaling?
- How the information presented in this section considered 'validation'?
- This section appears to be missing a description of how the velocity spectra are calculated using the Lagrangian particle data.
- Is the statement on Lines 717-718 meant to imply that the averaging time is insufficient? If so, why present these results?

**References**

Bernardin, F., M. Bossy, C. Chauvin, P. Drobinski, A. Rosseau, and T. Salameh (2009). Stochastic downscaling method: application to wind refinement. *Stoch. Environ. Res. Risk Assess. 23*, 851–859.

---

## Referee Report (RR2)

**Review of acm-2015-1015**

**Overall Comments:**

This review concerns the manuscript acm-2015-1015, which I reviewed two times previously. I am still not in agreement with the authors on several aspects of the work. However, I do not think that agreement is necessary for publication. Given the back-and-forth with the authors along with the latest revision of the manuscript, I can at least determine what was done and that it appears to be consistent. Although my own personal opinion is that there are currently better ways for doing the type of downscaling that is the focus of this work, it will be up to future workers to decide what will ultimately be adopted and prevalent. My recommendation is to accept the work pending the authors' resolution of a few lingering issues (below).

**Previous Review Major Comments:**

1. Sub-grid turbulence model: I think we are starting to get through some of the confusion regarding this point, which in my opinion was driven by the combination of terminology used and a scattered description of what was done.

I will attempt to describe the model as briefly as possible. If my understanding is still incorrect, the authors should consider how they can edit the text such that the description is clear for future readers.

Let's consider only the equation for $V$:

$$V_{k+1} = V_k - \nabla_x \overline{p} \delta t - C_1 \frac{\varepsilon_k}{K_k} \left[ V_k - \langle V \rangle \right] \delta t + \sqrt{C_0 \varepsilon_k} \Delta B_{k+1}. \tag{1}$$

| Variable | How specified | Reference |
|----------|---------------|-----------|
| $V_{k+1}$ | Eq. 1 above | Line 374.5 |
| $V_k$ | particle velocity at previous iteration or initial condition | $k$ subscripts not defined? |
| $\nabla_x \overline{p}$ | Meso-NH coarse grid (forcing) | Lines 388-393 |
| $\varepsilon_k$ | Meso-NH coarse grid (forcing) | Lines 388-393 |
| $K_k$ | particle velocity along with average operator $\langle \cdot \rangle$ | Lines 441-443 |
| $\langle V \rangle$ | particle velocity along with average operator $\langle \cdot \rangle$ | Lines 366-368,441-443 |

Some final remarks regarding this point:

- Although I now think I understand what was done, and agree that it seems to be consistent, it still seems strange to me to dump the resolved velocity at the grid scale. The resolved velocity already resolves most of the TKE that the model is trying to replicate. Based on Fig. 1 of the reply, I have a hard time believing that the so-called 'velocity increment' approach was implemented correctly ($V$ looks ok, something is clearly wrong with $U$ and $W$ because if filtered it would not follow the trend of the Meso-NH velocity). Regardless, the authors are free to use any approach they like, and it will be up to the community to decide which methodologies are ultimately adopted.

- The description of the methodology in the paper is, in the opinion of this reviewer, difficult to follow and cumbersome. Just to figure out how to implement Eq. 1 (above) requires sorting through several hundred of lines of text. I would think that a much clearer and condensed description is possible.

2.-5. Others: We are either in agreement or at an impasse regarding the other major comments, and no further discussion is likely to be fruitful.

**Minor Comments from Previous Review:**

16. Sect. 6.1.3 (velocity spectra) : I have quite a few concerns with this section : – For the SLM used here, we know that 1. An ensemble of particles at any location should have TKE K, and 2. The variance of particle velocity increments at any location should be $\langle du^2 \rangle = C_0 \varepsilon dt$, which is consistent with Kolmogorov theory. Given this, should we expect the particle velocity spectra to follow $k^{-5/3}$ scaling?

*The SLM has been designed to follow the $k^{-5/3}$ scaling [6]. Here the idea is to assess the behaviour of particles driven by the SLM when it is forced by Meso-NH ?that is to say when the pressure gradient and the dissipation rate are given by Meso-NH.*

I have searched the references to the work of Pope, and have found no mention of a requirement that the model should follow $k^{-5/3}$ scaling, rather that it should follow the second similarity hypothesis of Kolmogorov which implies $\langle du^2 \rangle = C_0 \varepsilon dt$. My understanding is that $k^{-5/3}$ is a consequence of both the first and second similarity hypotheses together. However, through our previous discussion we agreed that the model satisfies only the second similarity hypothesis. Can the authors please provide a more specific reference to the work of Pope to help me understand why we should expect the model to follow $k^{-5/3}$ scaling? If not, it seems that Sect. 6.1.3 requires some revision.

---

## Author Response (AR2)

First we would like to thank the reviewer for the second review which is full of ideas. They have been studied carefully but they have not been applied yet due to the framework which is not as free as we wish. In addition to the computational cost of particle modeling, we underline that the Meso-NH simulations have been performed for the BLLAST experiment and cannot be tuned at will.

Below is our response to the comments (in blue) on a point-by-point basis. The text referring to the article is indicated in italic.

**Answers to major comments**

**1 Sub-grid turbulence model**  My previous comment was essentially that it appears that the model is inconsistent with behavior that is expected in a subgrid turbulence model. My claim was that the particles should be driven by the resolved Eulerian velocity field, and the Lagrangian model should be predicting the unresolved velocity.

Based on my understanding, the behavior of the model presented by the authors appears to be inconsistent with that of a subgrid-scale model. Perhaps this is because my understanding of the model formulation is still incomplete, in which case the authors may be able to explain its consistency through the following simple thought experiment. Simply consider how the model should respond as grid resolution is varied. As the grid scale approaches the Kolmogorov scale, the subgrid effects should tend toward zero (this is the most fundamental quality of a subgrid model). And in the context of the particle model, the particle velocity should match the Eulerian velocity : $V \to v$ and $W \to w$. Forgive me, but looking at the equation following Line 419 I don't see how this will happen. In the opinion of this reviewer, the authors either need to demonstrate that the model is at least consistent in this regard, or use a model formulation that is consistent.

I am still not entirely sure why the authors have set up their Lagrangian equations the way they have, rather than using a standard approach that is known to be consistent (it is no more difficult or costly). The authors are of course free to use any approach they like as long as it's consistent, I am just curious.

So why not calculate the total velocity $V$ as the sum of the resolved Eulerian grid velocity (known, call it v) and model the unresolved component using the SLM (unknown, call it $\tilde{v}$) ; i.e., $V = v + \tilde{v}$ ? Note that in the current paper, $v = -\nabla_x p$ which is wrapped into the total equation for $V$, whereas here $v$ is interpolated from the Eulerian simulation to the particle location and thus is 'known'. Then the evolution equation for $v$ is left to model, which is

$$dv\tilde{} = -\, C_1 \frac{\varepsilon}{e}\, \tilde{v}\, dt + \, \sqrt{C_0 . \varepsilon}\, dB \tag{1}$$

where $\varepsilon$ and $e$ are turbulent dissipation rate and subgrid TKE, respectively, which are interpolated from the Eulerian grid to the particle position. Now, our model will at least be consistent as the grid is refined : $e$ tends toward zero as the grid scale tends toward zero (as a result this dissipation term becomes very large and damps out all fluctuations), and thus $\tilde{v} \to 0$ and $V \to v$. This formulation is also consistent if you go the other way and tend toward RANS. As the grid scale becomes very large, $e \to K$, $v \to \overline{v}$, $\tilde{v} \to v'$ , and thus we converge to a standard RANS downscaling

model. This seems important for Meso-NH, which can be 'switched' between RANS and LES modes. Only in the case of RANS where no turbulence is resolved is the resolved velocity equal to $-\nabla_x p$ and the unresolved TKE equal to $K$.

An important point has to be clarified : the SLM aims at modeling the total wind speed using particles which sample the local wind pdf. The SLM has been designed regardless of the grid size of the model used to force it. Nevertheless the term "local" is related to the forcing scale. It can be understood as " a smaller scale than the forcing scale". Thus, if the forcing scale tends toward zero, the particles would sample the wind pdf into a very small of atmosphere. There is one limit : the SLM should be used to model wind and turbulence in the inertial sub-range.

The stochastic Lagrangian model that we used is consistent with the model described by Pope [6, 7]. The equation for the horizontal velocity $V$ is as following :

$$V_{k+1} = V_k - \nabla_x \overline{p} \, \delta t \; - \; C_1 \frac{\varepsilon_k}{K_k} \left[ V_k - <V> \right] \delta t + \; \sqrt{C_0 . \varepsilon_k} \; \Delta B_{k+1}^V$$

and the equation for the velocity increment is simply given by :

$$dV_{k+1} = -\nabla_x \overline{p} \, \delta t \; - \; C_1 \frac{\varepsilon_k}{K_k} \left[ V_k - <V> \right] \delta t + \; \sqrt{C_0 . \varepsilon_k} \; \Delta B_{k+1}^V \qquad (2)$$

where $K_k$ is the turbulent kinetic energy and $\varepsilon_k$ the dissipation rate. The average $< \cdot >$ is computed using subsets of the particle system and a regularization function. The pressure gradient $-\nabla_x \overline{p} \, \delta t$ represents the increment of the averaged horizontal velocity. All these terms are also described by Pope in [6] for instance.

Using $V = v - \tilde{v}$, the terms of the equation 1 can be rearranged as following :

$$dV = dv - \; C_1 \frac{\varepsilon}{e} \left( V - v \right) dt + \; \sqrt{C_0 . \varepsilon} \; dB \qquad (3)$$

where $dv$ is the increment of the horizontal Eulerian velocity, $e$ the subgrid TKE and $\varepsilon$ the dissipation rate. Thus there are two differences between the equations 2 and 3. The first one is the term chosen for the averaged velocity increment. We have chosen to use the pressure gradient to be consistent with the model described by Pope, whereas you suggest to use directly a velocity increment. In our work the model is coupled with a grid point model and the pressure gradient is computed using the grid point pressure field. If the grid scale approaches the Kolmogorov scale, there might be difficulties to compute a local pressure gradient. Thus in this case, our choice is not relevant, and the velocity increment should be used.

The second difference is about the TKE. In equation 2, the TKE $K$ is the total TKE computed using the particle velocity. In equation 3, the sub-grid TKE $e$ is used. This formulation is not consistent with the model of Pope which aims at modeling the total velocity. This is the main difference between the two equations.

To conclude, another choice can be done for the averaged velocity increment. A sentence will be added in the article to precise our choice. A quick test has been done by forcing the particle with the velocity increment. The results seem poorer (see figure 1).

Lines 445 of the authors' response : *The pressure gradient has be chosen to be consistent with the model described by Pope [7], but the mean horizontal velocity can be used instead in case of difficulty in computing the pressure gradient.*

[Figure]

Fig. 1: Time evolution of the three wind components obtained from the fine Meso-NH simulation (black) and by the particle model (red) for one cell of the fine grid when the particle are forced directly by the velocity increment

**2 Gaussian assumption**    Unfortunately, I am still in disagreement with the authors' position, as well as their newly added statements, e.g., "It leads that in this study, in a given grid cell, particles are samples of different Gaussian pdf. " It doesn't matter if the turbulence statistics vary in space, the particle velocities will be locally Gaussian with variance, dissipation, etc. equal to the Eulerian value specified at that point. This is the idea behind the Pope/Thomson Lagrangian stochastic dispersion models. We specify a PDF at every point, and we seek to generate an ensemble of Lagrangian particles that has that PDF and has a local dissipation rate of $\varepsilon$. This is how the models are derived, and how they work out in practice. Any deviation from this is caused by numerical error (or in the authors' case it could be due to the arbitrary stochastic terms that were added). The advantage of using an SLM rather than just sampling a PDF is that the particle velocities are correlated in time (this is a result of the fact that they must have dissipation rate of $\varepsilon$, or more directly that they follow Kolmogorov's second similarity hypothesis). Also as a side note regarding why the histograms presented by the authors (Fig. 1 of reply) do not appear Gaussian : the sample size is too small to make such an assessment. With so few samples (i.e., 75), the histograms are unlikely to look Gaussian unless we get lucky. The authors can test this using the MATLAB code I provided in the previous review, and setting N=75. In this case the PDF will usually look quite non-Gaussian (depending on 'luck'), and as N is increased the PDF converges toward Gaussian.

With all that said, I am not particularly concerned with this issue as it pertains to the manuscript. The original motivation for this comment was related to the novelty of the paper. Let's say for the sake of argument that the authors are correct that for some reason the particle velocities are highly non-Gaussian when aggregated over a grid cell. Even if that is so, what is the novelty of the downscaling methodology when considering the work of, e.g., Bernardin et al. (2009) ? Looking at their Eq. 19b, the only difference I see between the authors' velocity equation is the arbitrary "buoyancy" term that was added. Based on my own viewpoint, I would say that the novelty lies in the fact that the authors have used the Meso-NH model, which based on my limited understanding, is somewhat of a hybrid between a RANS and LES code. In that case, it seems that the downscaling method should be consistent in that regard, and thus it might make more sense to use the approach described in the previous comment.

As written in the previous answer the stochastic Lagrangian model (equations lines 419 of the author's response) describes actually a Gaussian pdf when it is applied to the whole particle system with the same forcing and when $< \cdot >$ is the ensemble average over the particle system. However in our work, the forcing varies

in space and the average $< \cdot >$ is computed for each particle using a different subset of the particle system and a regularization function. Thus a particle is driven by its own model and the particles are not identically distributed. In a grid cell, it leads to a mixing of local Gaussian models.

The buoyancy term in our Lagrangian model is not a significant term in this work. The buoyancy term indicates only that the model can be improved by modeling the temperature. In our work, it is simply modeled by a centered Gaussian variable with a very small variance.

There are several major differences between our method and the one described by Benardin in [3] :

- in [3] the forcing is done by giving boundary conditions for the particle velocities whereas in our work particles are forced by the pressure gradient and the dissipation rate in each cell,

- the boundary is "solid" in [3], that is to say that the particles rebound against the boundary, whereas we have chosen to let particles follow the air flow and to replace outgoing particles by new particles,

- Bernardin uses a $k - \varepsilon$ closure scheme for the dissipation rate whereas the dissipation rate is given by the grid-point model in our work,

- the way to compute the average $< . >$ differs : in our work it is computed using a Gaussian regularization function whereas in [3] it is the average of the particle velocities cell by cell – that is to say using a characteristic function. The advantage of our method is to take into account the continuity of the modeled atmosphere,

- the modeled scales are different : the horizontal grid size is 3km large in [3] whereas it is 40m in our work. Thus the application of these works are complementary but different.

**3 Stochastic numerical error 'correction' term in the particle position equation**

Can the authors provide any references that 'correct' for the numerical errors in this way ? I can't see how adding additional dispersion somehow corrects for the numerical dispersion – in this case I don't see how two wrongs make a right, as both the error and correction appear to be additive and dispersive. Typically, numerical truncation errors are reduced by reducing the timestep, or there are refinement methods that use a systematic approach to (usually iteratively) improve the solution. Maybe the authors are using some method that I am unfamiliar with, in which case I am curious to read more (perhaps from a reference). Also, I would note that the 'correction' is being added to the position evolution equation, yet the explanation given by the authors seems to pertain to errors related to the velocity evolution equation. How are the two related ?

The referee points out an interesting remark on the dispersive character of a noise added on the particle position. Let's back to the formulation.
The exact system is in continuous time without any noise added on the particle location. The Lagrangian particle velocity model is of McKean-Vlasov type and a Euler-Maruyama scheme is used. Some works of Bally and Talay (ex : [?]) give the shape of the integration errors for SDE. Then denoting $V_{n-1}$ the velocity at time $t_{n-1}$, $U_n$ the velocity at the time $t_n$ given by the continuous integration starting

from $V_{n-1}$ and $V_n$ the approximated velocity given by the Euler integration scheme. Finally the Euler scheme error is denoted by $v'_n$ and $U_n = V_n + v'_n$. $v'_n$ is a random variable and the paper [2] gives information on its probability laws.

Concerning the integration of the particle locations, it yield with the same kind of notations, $X_{n+1} = X_n + \int_{t_{n-1}}^{t_n} U_s ds$. Then using the Euler approximated integration, the particle position is a random variable integrating the random variable $v'_n$ along the path. This random path integration gives a random variable (see [5]). As suggested by the referee, it is possible to proceed to the approximation using small timesteps. But it always introduce an integration errors. Then to model this integration error, a random variable has been retained. Historically it comes from our works [1] on nonlinear filtering of velocity measurements on turbulent flows. Using the Bally and Talay work, using the shape of the particle Lagrangian model which is locally Gaussian, a Gaussian modeling seems to be suitable. There is an interesting discussion (section on particle mesh method and section on application) on this topic in the article of Pope [8]. Formally it is one of the reason that the discrete time nonlinear filtering problem solved by particle approximation do not collapse on a singularity.

In this work the limitation of the numerical costs has been privileged. Then the path errors due to the discrete time integration has been modeled by a Gaussian random variable. The variance of this Gaussian noise is limited to get a small random term with respect to the approximated velocity term. As remarked by the referee this choice of a random variable is dispersive. With a small number of particles, the dispersive effect isn't a drawback and allow to the particles to visit all the domain.

**4 Rogue trajectories**   Overall, the authors' response was sufficient. I would recommend one thing the authors might try. What happens to the power spectra when you decrease the particle integration timestep $\delta t$ by, say, an order of magnitude ? Does it get rid of some of that extra energy at small-scales ? If so, there may be some issues with numerical stability although it might not be manifesting as 'rogue' trajectories in the traditional sense.

It is an interesting idea, but decreasing the particle integration timestep by an order of magnitude increases the computational time by the same order. In view of our numerical framework, it is not possible to decrease the timestep without using another programming language.

**5 Validation**   Even on the "fine" Meso-NH grid, the velocity field is presumably still missing unresolved (subgrid) energy, which I'm guessing is not negligible (looking at the spectra in Fig. 9, it appears to be substantial). This is why the Mesh-NH velocity is much "smoother" than the particle velocity. So the When you filter (i.e., average) to get the "coarse" grid, some additional energy is removed, let's call that $\Delta e$. So the total TKE for the coarse grid is $K_{res} + \Delta e + e$. Here is something to try : why not use $\Delta e$ instead of $e$ in Eq. 1 (above), and compare to the resolved TKE from the "fine" solution ? In this case, the goal of the downscaling model is to

recover $\Delta e$ rather than $\Delta e + e$, which means you can directly compare to $K_{res}$ for the fine grid. My opinion is that simply saying that for future work "higher resolution simulations should be performed" is not acceptable.

Regarding a "toy problem" : I could think of some tests that could be useful here. What about generating some random 'turbulence' (could be white noise or correlated) on the "fine" grid, then filtering it to get a "coarse" grid? You could calculate the TKE of the fine grid turbulence (this is '$K = K_{res} + e$'), then calculate the TKE of the coarse grid (this is $K_{res}$). Now drive the particle model with e and downscale to the fine grid, where you should find that the TKE of the total particle velocity is K. This of course is non-physical and probably wouldn't go into the paper, but could be a good verification check to demonstrate consistency.

We underline that, in equation 2, $K$ is the total TKE modeled by the particles. Using the subgrid TKE as suggested would not be consistent with the Lagrangian model described by Pope.

The idea behind the sentence "higher resolution simulations should be performed" is to perform grid point simulations with a finer grid size without changing the forcing scale. Thus the idea is to compare the particle fields that we already have to grid point simulations using a grid finer than 40m x 40m x 12m. Unfortunately, it has not been possible to perform such simulations with Meso-NH for this work.

The suggested test does not seem suitable with our forcing method. The subgrid TKE is not use to force the particle system. Indeed, we use the total particle TKE in the Lagrangian model and the pressure gradient and the dissipation rate are used to force the particle system. Forcing the particle system by the TKE requires a new formulation of the model. Thus we have not adapted the test to our work, but it might be done in further works.

**Answer to minor comments :**

1. In my own experience, the term 'downscaling' is usually used to describe a one-way model from large to small scales, whereas 'sub-grid modeling' is typically reserved for two-way coupling where the large-scale model needs to parameterize the small-scales. I would consider this work to address downscaling. I would leave it up to the authors discretion, but they may consider revising the title and certain other instances to make this point clear.
   Thank you for the suggestion. It has been changed in the manuscript.

2. Line 25 : model → models
   The correction has been made.

3. Lines 41-44 : Consider re-wording this sentence. How can AROME airport resolve processes? The authors probably mean that processes at the scale of AROME airport are not resolved.
   The correction has been made.

4. Lines 80-86 : The first statement seems to say that an assumption is made that the local PDF is Gaussian. Then it says that locally the PDF samples multiple Gaussian PDFs, and therefore it is not necessarily Gaussian. These seem to contract each other.
   We suggest to add a sentence to clarify this point. Line 70 of the previously

revised manuscript / line 80-86 of the authors' response : *The method we suggest differs from these previous works : the Gaussian assumption on the pdf shape is only locally made. This locally Gaussian assumption is linked to the use of a local average operator presented in section 3.6. As explained in this section, in this study the locally Gaussian assumption is not equivalent to have a Gaussian pdf in each cell. It leads that in a given grid cell, particles are samples of different Gaussian pdf. Therefore they give access to a discrete pdf which is not necessarily Gaussian.*

5. Sect. 3.2 : What is meant by the term 'coupling experience'? In English, this phrase sounds a bit unusual. Is there some particular reason to use the word 'experience', rather than just saying something like 'model coupling', or 'coupling between resolved and unresolved scales'?
   The correction has been made.

6. Sect. 3.3 : Consider moving this section until after the model has been introduced (i.e., beginning of Sect. 6). Currently, it feels out of place since this is really just details related to model testing/validation and is not central to the model itself.
   The section 3.3 has been moved in section 4.

7. 400-403 : Can the authors explain why they feel that the ideas presented by Kolmogorov are considered "laws"? Typically these ideas are referred to as 'Kolmogorov theory' or 'Kolmogorov's hypothesis', as they are largely based on similarity/scaling arguments.
   In French, the word "laws" is used. It has been replaced by "theory" in the manuscript.

8. 400-403 : How exactly is the model consistent with Kolmogorov theory? Would the authors consider it to be consistent with all of the similarity hypotheses presented in K41, or is it that it is consistent with Kolmogorov's second similarity hypothesis in that the variance of the Lagrangian velocity increments ($< du^2 >$) is proportional to the turbulence dissipation rate, i.e., $C_0 \varepsilon dt$?
   As explain by Pope in [6, 7], the model is consistent with Kolmogorov's second similarity hypothesis. This point has been added to the manuscript.

9. Lines 419.5 (equation) : I am not entirely clear on how K is specified. Normally, this would come from the large-scale simulation and be interpolated to the particle position. On Lines 308-310, the authors mention that the TKE is extracted from Meso-NH, which would lead me to believe that is where K comes from. However, Sect. 5.2 would suggest otherwise, that somehow the TKE is calculated afterword, although it is required in the velocity equation itself. Please explain.
   As it is explained lines 447-451 of the authors' response, the term $K$ in the SLM equations does not come from the model Meso-NH. This variable represents the total TKE and it is not used to force the particle system. It is computed using the particle velocities and a local average operator. This kind of average operator is also used by Pope in [7].

10. Lines 423-426 : The statement regarding the buoyancy term is vague. There are an infinite number of ways in which this term could be modeled using a random variable. Was this simply an empirical 'knob' that was turned?
    The buoyancy is modeled by a centered Gaussian variable with a small variance

of 0.05 $m^2/s^2$. The variance value has been set to a small value on purpose. Indeed the buoyancy term is in the model only to show the possibility to improve it by modeling the temperature.

11. Lines 504-505 : How is the particle velocity initialized ? This seems important considering that the particle simulation times ($\Delta t$) are much shorter than the integral time scale, and therefore they are likely to 'remember' the initial condition.
    There is an error lines 504-505 of the authors' response. The particle velocities are initialized using the velocities of the coarse Meso-NH simulation. The correction has been made in the manuscript.

12. Sect 3.6 : This title seems inappropriate. When are structure functions ever calculated in the paper ? It seems like a more appropriate title might be 'Ensemble averaging' or something of that nature.
    The title of section 3.6 has been changed as suggested.

13. Lines 635-651 and Figure 7 : I don't see the value of this comparison. Firstly, the simulations and the data are not over the same time period. Secondly, am I supposed to look at Fig. 6 and Fig. 7 and say "Yes, the Meso-NH velocity is smoother than the sonic data" ? It would be incredibly surprising if that were not the case considering that the simulations dont resolve below the grid scale.
    In agreement with the comment, the figure 7 has been removed. A reference to [?] has been added.

14. Lines 671-674 : Should we not expect better agreement if the method is consistent ?
    We continue to find that the results are in good agreement with the fine Meso-NH simulations, especially for the horizontal wind. It leads us to think that the forcing method works but the Lagrangian model may be improved for the vertical velocity. The following lines have been added (line 668) : *However one can notice that the results are better for the horizontal wind components than for the vertical one. To improve this point, one way could be to supplement the SLM with an equation for the temperature in order to model the buoyancy effect.*

15. Lines 855-858 and Fig. 12 caption : It is not clear to me how exactly the 'new' grid was obtained. If the grid resolution is changed, shouldn't the 'Meso-NH' TKE (black line) change as well ?
    The new coarse grid is obtained by averaging the fine grid by groups of 2x2x2 cells instead of 4x4x2 cells for the previous coarse grid. Thus the new grid size is 40mx40mx24m. In the figure showing the different TKE (figure 12 of the first authors' response), the black line represents the TKE of the fine Meso-NH simulations. The resolution of these simulations does not vary in our work. The following clarification for the new coarse grid has been added line 857 : *This new grid is obtained by averaging the fine grid on 2x2x2 cells.*

16. Sect. 6.1.3 (velocity spectra) : I have quite a few concerns with this section :
    - For the SLM used here, we know that 1. An ensemble of particles at any location should have TKE $K$, and 2. The variance of particle velocity increments at any location should be $< du^2 >= C_0 \varepsilon dt$, which is consistent with Kolmogorov theory. Given this, should we expect the particle velocity spectra to follow $k^{5/3}$ scaling ?

The SLM has been designed to follow the $k^{-5/3}$ scaling [6]. Here the idea is to assess the behaviour of particles driven by the SLM when it is forced by Meso-NH –that is to say when the pressure gradient and the dissipation rate are given by Meso-NH.

- How the information presented in this section considered 'validation'?

As the modeled scales are in the inertial subrange, the wind spectra are expected to follow the $k^{-5/3}$ scaling. This section aims at ensuring that the particle wind spectra follow the scaling.

- This section appears to be missing a description of how the velocity spectra are calculated using the Lagrangian particle data.

First the particle velocities are averaged fine grid cell by fin grid cell in order to get a grid point field for the particle velocity. Then the particle velocity spectra are calculated using the same method than for the Meso-NH velocity spectra. The explanation is given lines 692-697 of the previous authors' response.

- Is the statement on Lines 717-718 meant to imply that the averaging time is insufficient? If so, why present these results?

These results show clearly that the particle wind spectra present regular slopes. This is a good point to assess the use of the stochastic downscaling method. The SLM is also used for turbulence estimation [9], and in this context, the spectra are computed using longer data sets – about 1h of data with a 4 second time step. This is why we think that longer simulations are needed to fully validate the model behaviour.

[revised manuscript text omitted]

---

## Author Response (AR3)

Once again, we thank the reviewers for the enriching reviews which have improved the quality of this article. Here are our answers to the last comments and remarks.

**Answer to final remarks**

Although I now think I understand what was done, and agree that it seems to be consistent, it still seems strange to me to dump the resolved velocity at the grid scale. The resolved velocity already resolves most of the TKE that the model is trying to replicate. Based on Fig. 1 of the reply, I have a hard time believing that the so-called 'velocity increment' approach was implemented correctly (V looks ok, something is clearly wrong with U and W because if filtered it would not follow the trend of the Meso-NH velocity). Regardless, the authors are free to use any approach they like, and it will be up to the community to decide which methodologies are ultimately adopted.

The description of the methodology in the paper is, in the opinion of this reviewer, difficult to follow and cumbersome. Just to figure out how to implement Eq. 1 (above) requires sorting through several hundred of lines of text. I would think that a much clearer and condensed description is possible.

The remarks have been carefully read, and we agree that the implementation of the Stochastic Lagrangian Model is not straightforward. The implementation requires to read several sections, but we think that all the given explanation are necessary.

Concerning the 'velocity increment' approach, the implementation is exactly the same for U, V and W. The code was checked when the results were obtained, and no error was found.

**Answer to Minor Comments from Previous Review**

I have searched the references to the work of Pope, and have found no mention of a requirement that the model should follow $k^{5/3}$ scaling, rather that it should follow the second similarity hypothesis of Kolmogorov which implies $< du^2 >= C_0 \varepsilon dt$. My understanding is that $k^{5/3}$ is a consequence of both the first and second similarity hypotheses together. However, through our previous discussion we agreed that the model satisfies only the second similarity hypothesis. Can the authors please provide a more specific reference to the work of Pope to help me understand why we should expect the model to follow $k^{5/3}$ scaling? If not, it seems that Sect. 6.1.3 requires some revision.

In [4], Pope presents the energy cascade and the Kolmogorov hypotheses in the chapter 6. The energy spectrum and the $-5/3$ law are described section 6.1.3. Note that, whatever they are called, the assumption needed to get the -5/3 spectrum is the following

$$E(l) = C_0(\varepsilon l)^{2/3} \tag{1}$$

with $l$ the characteristic size of the eddy of kinetic energy $E(l)$. The rate of transfer from different size $\varepsilon$ is assumed to not depend on the size (see [2, 1]). By taking the

Fourier transform of 1 and with the substitution $\alpha = kl$, we get the -5/3 spectrum :

$$
\begin{aligned}
E(k) &= \int_0^{+\infty} E(l) e^{-ilk} dl \\
&= \int_0^{+\infty} C_0 (\varepsilon l)^{2/3} e^{-ilk} dl \\
&= C_0 \varepsilon^{2/3} \int_0^{+\infty} \left(\frac{\alpha}{k}\right)^{2/3} e^{-i\alpha} \frac{d\alpha}{k} \\
&= C_0' \varepsilon^{2/3} k^{-5/3}
\end{aligned}
\tag{2}
$$

This equivalence between equation 1 and the -5/3 spectrum is stated in [1] (equation 2.10 and 2.11) and in [3] (equation 9 and 13 are said "exactly equivalent" in the paragraph after equation 13).

Manuscript prepared for Atmos. Chem. Phys.
with version 2015/09/17 7.94 Copernicus papers of the LaTeX class copernicus.cls.
Date: 28 March 2017

**A new downscaling method for sub-grid turbulence modeling**

**L. Rottner[1], C. Baehr[1], F. Couvreux[1], G. Canut[1], and T. Rieutord[1]**

[1]Météo-France - CNRS, CNRM / GAME, UMR 3589, 42 avenue Coriolis, 31100 Toulouse

*Correspondence to:* Lucie Rottner (lucie.rottner@meteo.fr)

**Abstract.**

In this study we explore a new way to model sub-grid turbulence using particle systems. The ability of particle systems to model  small-scale turbulence is evaluated using  high-resolution numerical simulations. These high-resolution  data are averaged to produce a coarse-grid velocity field which is then used to drive a complete particle-system-based downscaling. Wind fluctuations and turbulent kinetic energy are compared between the particle simulations and the high-resolution simulation. Despite the simplicity of the physical model used to drive the particles, the results show that particle system is able to represent the average field. It is shown that this system is able to reproduce much finer turbulent structures than the numerical high-resolution simulations. In addition, this study provides an estimate of the effective spatial and temporal resolution of the numerical models. This highlights the need for higher resolution simulations  in order to evaluate the very-fine turbulent structures predicted by the particle systems.  Finally, a study of the influence of the forcing scale on the particle system is presented.

**1 Introduction**

Following the increase in computing power, the resolutions of meteorological models have increased steadily over the past years. The refinement of the temporal and spatial resolution of atmospheric  models requires a finer and finer representation of physical phenomena. The current weather forecast models have  resolutions of approximately one kilometer. However, the small processes, which have local effects, are still sub-grid processes in such models. Thus, they are subject to physical parametrization.

The  issue of downscaling concerns many meteorological research fields, from snow pack modeling to  cloud-cover modeling. A particularly delicate matter is  the modeling of the turbulence in the Atmospheric Boundary Layer (ABL). In the ABL, there is a transfer of energy from  scales of the order of a kilometer down to sub-meter scales. This transfer is called the energy cascade. Thus, whatever the model resolution, some turbulent processes are sub-grid processes. For numerical weather forecast models, the processes associated to  sub-kilometer scales are not resolved yet. For instance, a recent study shows that these processes are not resolved at the scale of AROME Airport which has a horizontal resolution of 500 meters (Hagelin et al., 2014).  Since turbulence is a  key driver of the evolution of local-scale atmosphere, it is critical that turbulence processes are parametrized well. For instance, recent studies have shown the influence of the turbulence parametrization on the cloud modeling in tropical regions (Machado and Chaboureau, 2014). Several field experiments have helped to understand the influence of  small-scale turbulence on local weather conditions – the erosion of the nocturnal valley inversion for instance (Rotach et al., 2004; Drobinski et al., 2007; Rotach et al., 2008).

Because of their variability and their sensitivity to local conditions, these turbulent phenomena are especially difficult to model. Instead of a  reduction in grid size, we suggest here another way to model sub-grid turbulence. In this paper, we present a stochastic downscaling approach. Our method is based on particle systems that are driven by a local turbulence model. Those particles are embedded in grid cells (illustration 1). From the mathematical point of view, the particles sample the probability density function (pdf) of the sub-grid wind. The descrip-

[Figure]

*Coarse Meso-NH grid*

$\Delta x_{forcing} = 160$ m

Forcing

Sub-grid contribution

$\Delta x_{particle} \simeq 1$ m

*Sub-grid particles*

$\Delta x_{reference} = 40$ m

*Reference simulation*

[revised manuscript text omitted]